# Amoeboid cells undergo durotaxis with soft end polarized NMIIA

**Chenlu Kang**[1,2,3†]**, Pengcheng Chen**[4†]**, Xin Yi**[1,2,3†]**, Dong Li**[5,6]**, Yiping Hu**[1,2,3]**, Yihong Yang**[5,7]**, Huaqing Cai**[5,7]**, Bo Li**[4]*****, Congying Wu**[1,2,3]*****

[1]Institute of Systems Biomedicine, School of Basic Medical Sciences, Peking University Health Science Center, Beijing, China; [2]Institute of Advanced Clinical Medicine, State Key Laboratory of Molecular Oncology, Beijing, China; [3]International Cancer Institute, Peking University, Beijing, China; [4]Institute of Biomechanics and Medical Engineering, Applied Mechanics Laboratory, Department of Engineering Mechanics, Tsinghua University, Beijing, China; [5]National Laboratory of Biomacromolecules, Institute of Biophysics, Chinese Academy of Sciences, Beijing, China; [6]School of Life Sciences and Medicine, University of Science and Technology of China, Hefei, China; [7]College of Life Sciences, University of Chinese Academy of Sciences, Beijing, China

*For correspondence:
libome@tsinghua.edu.cn (BL);
congyingwu@hsc.pku.edu.cn
(CW)

†These authors contributed
equally to this work

Competing interest: The authors
declare that no competing
interests exist.

Reviewing Editor: Yi Arial Zeng,
Chinese Academy of Sciences,
China

## eLife Assessment

This study presents an **important** finding on durotaxis in various amoeboid cells that is independent of focal adhesions. The evidence supporting the authors' claims is **compelling**. The work will be of interest to cell biologists and biophysicists working on rigidity sensing, the cytoskeleton, and cell migration.

**Abstract** Cell migration towards stiff substrates has been coined as durotaxis and implicated in development, wound healing, and cancer, where complex interplays between immune and non-immune cells are present. Compared to the emerging mechanisms underlying the strongly adhesive mesenchymal durotaxis, little is known about whether immune cells - migrating in amoeboid mode - could follow mechanical cues. Here, we develop an imaging-based confined migration device with a stiffness gradient. By tracking live cell trajectory and analyzing the directionality of T cells and neutrophils, we observe that amoeboid cells can durotax. We further delineate the underlying mechanism to involve non-muscle myosin IIA (NMIIA) polarization towards the soft-matrix-side but may not require differential actin flow up- or down-stiffness gradient. Using the protista *Dictyostelium*, we demonstrate the evolutionary conservation of amoeboid durotaxis. Finally, these experimental phenomena are theoretically captured by an active gel model capable of mechanosensing. Collectively, these results may shed new lights on immune surveillance and recently identified confined migration of cancer cells, within the mechanically inhomogeneous tumor microenvironment or the inflamed fibrotic tissues.

## Introduction

Cell migration is essential to myriads of physiological and pathological processes including cancer dissemination (*Trepat et al., 2012*). As the mechanical properties of tumor microenvironment are gradually unveiled, waves of studies have focused on the contribution of mechanical force - in forms of proliferation generated cell jamming (*Kindberg et al., 2020*) and fibrosis-induced tissue stiffening

- on tumor expansion as an well as invasion (*Wells, 2013*). The ability of cells to sense and navigate along higher substrate stiffness has been coined as 'durotaxis' in the early 2000s (*Lo et al., 2000*) with important pathophysiological implications (*Espina et al., 2022*). Over the past two decades, accumulating in vitro and in vivo evidence has demonstrated durotaxis in tumor epithelial sheets and single mesenchymal cells (*Raab et al., 2012*; *DuChez et al., 2019*; *Sunyer and Trepat, 2020*; *Shellard and Mayor, 2021a*; *Shellard and Mayor, 2021b*), and revealed mechanisms including focal adhesion tugging or probing of the local substrate by actin-based protrusions to underlie this specialized directional cell migration (*Raab et al., 2012*; *Sunyer et al., 2016*; *Plotnikov and Waterman, 2013*; *Plotnikov et al., 2012*). However, compared to tumor cells that harness the capacity to durotaxis, little is known about whether microenvironmental stiffness differences could determine the destination of immune cells.

In contrast to mesenchymal migration adopted by some tumor cells, immune cells mainly undergo rapid single-cell crawling referred to as 'amoeboid' migration, without the formation of stable focal adhesions but can achieve migration ten times faster (*Shannon et al., 2019*). Modeling on amoeboid migration suggested that myosin contractility may play an important role when mechanical anchoring is less evident (*Hawkins et al., 2011*). In this model, actomyosin was described as an active gel whose polarity dictates the alignment of actin fibers (*Hawkins et al., 2011*). Moreover, in silico simulation suggested that actomyosin contractility can lead to collective internal flows of actin, which, in turn, propels the cell body forward (*Tjhung et al., 2012*). Meanwhile, experiments have also revealed that protrusive flowing of the anterior actin network drives the adhesion-independent cell migration.

During development and immune surveillance, chemotactic cues have been reported to globally prescribe the migratory direction and destination of immune cells (*Carles and Pittet, 2010*). Using mammalian leukocytes and lymphocytes (*Stuelten et al., 2018*; *Wu et al., 2012*) as well as the protista *Dictyostelium* (*Manahan et al., 2004*), mechanisms of amoeboid chemotaxis have been extensively studied. Instead, little is known about whether these amoeboid cells would undergo durotactic migration. Previous studies have shown that inflamed, stiffened endothelial cells could induce more neutrophil transmigration (*Stroka and Aranda-Espinoza, 2011*), and that once adhered to the endothelium, transmigrating neutrophils may exert durotactic sensing to search for stiffer sites (*Schaefer and Hordijk, 2015*). In addition, lymphocytes such as CD4$^+$ Naïve T cells increased spreading area when plated on stiffer substrates (*Saitakis et al., 2017*), indicative of the capacity to sense external mechanical signal. The above evidence argue that amoeboid cells, albeit without strong adhesions, are able to sense and respond to environmental stiffness.

Here, we developed an imaging-based cell migration device that provided both spatial confinement and stiffness gradient. By tracking live cell trajectory and analyzing the directionality of CD4$^+$ Naïve T cells and neutrophils, we observed that amoeboid cells can durotaxis. We further uncovered the underlying mechanism to be NMII dependent. Based on our observations that NMIIA was polarized towards the soft-end in cells on stiffness gradient gel, we mathematically simulated amoeboid cells as active droplets, predicting that durotaxis can be strengthened on the softer region of the gradient substrate which was then verified experimentally. Moreover, we demonstrated the evolutionary conservation of amoeboid migration in the protista *Dictyostelium*. Together, the identification that rigidity-guided migration existed in low-adhesive, fast-moving amoeboid cells may provide new insights in focal adhesion-independent mechanosensing, and may shed new lights on immune surveillance and cancer metastasis.

## Results

### T cells and neutrophils undergo durotactic migration

T cells and neutrophils exert amoeboid migration in vivo and in vitro under confinement (*Hons et al., 2018*; *Mihlan et al., 2022*). To probe whether amoeboid cells would undergo durotaxis, we manufactured an imaging-based cell migration device which provided spatial confinement to maintain amoeboid cell migration (*Hons et al., 2018*; *Litschko et al., 2018*; *Abbasi et al., 2023*) and stiffness gradient to provide durotactic cue (*Figure 1A*). Typically, cells were sandwiched between two polyacrylamide

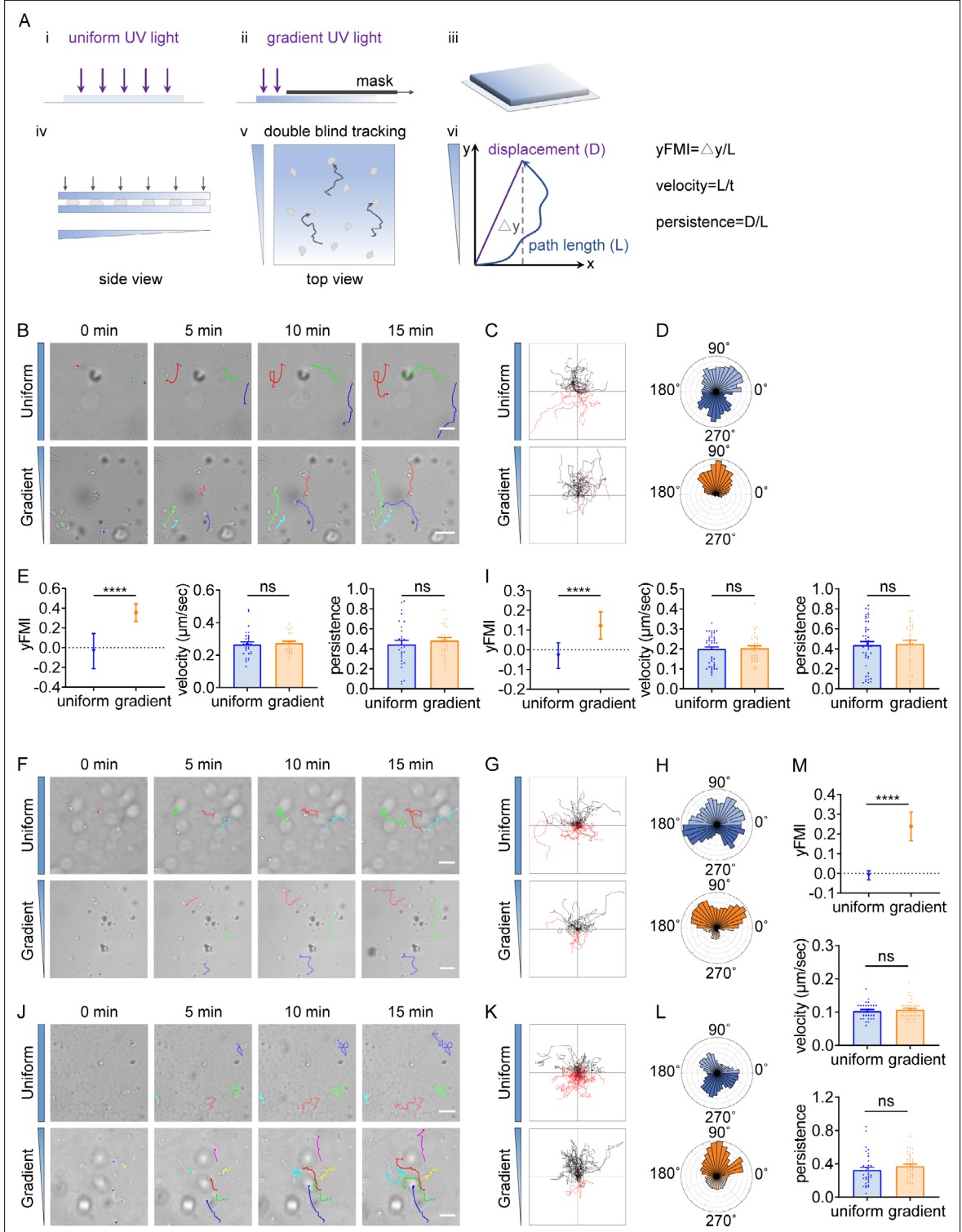

**Figure 1.** T cells and neutrophils undergo durotactic migration. (**A**) Illustration of the imaging-based cell migration device and yFMI value calculation. (**B**) Representative time-lapse images of single CD4[+] Naïve T cells migrating on Intercellular Adhesion Molecule 1 (ICAM-1) coated polyacrylamide gels with uniform stiffness (top, 30 kPa) and stiffness gradient (bottom, with greater stiffness towards the top, 20.77 kPa/mm). Color lines indicated the migration trajectories of a single cell. Scale bar: 50 μm. (**C**) Representative tracks of migrating CD4[+] Naïve T cells on the uniform stiffness gels (top)

*Figure 1 continued on next page*

*Figure 1 continued*

or stiffness gradient gels (bottom, with greater stiffness towards the top). Note that a higher proportion of cells migrate toward the stiffer end of the gradient substrate (black tracks) compared to cells migrating opposite to the gradient (red tracks). Tracks on the uniform substrate showed equal proportions for each part. (**D**) Angular displacement of CD4+ Naïve T cells on the uniform stiffness gels (top) or stiffness gradient gels (bottom, with greater stiffness towards the top). (**E**) y-FMI (Forward Migration Index), velocity and migration persistence of CD4+ Naïve T cells cultured on uniform substrate or gradient substrate (n≥30 tracks were analyzed for each experiment, N=3 independent experiments for each condition, replicates are biological). All error bars are SEM. ****p<0.0001, ns, not significant, by Student's *t*-test. (**F**) Representative time-lapse images of single neutrophils derived from mouse bone marrow migrating on fibronectin-coated polyacrylamide gels with uniform stiffness (top) or stiffness gradient (bottom). Color lines indicated the migration trajectories of a single cell. Scale bar: 50 µm. (**G**) Representative tracks of migrating neutrophils on the uniform stiffness gels (top) or stiffness gradient gels (bottom, with greater stiffness towards the top). (**H**) Angular displacement of neutrophils on the uniform stiffness gels (top) or stiffness gradient gels (bottom, with greater stiffness towards the top). (**I**) y-FMI, velocity, and migration persistence of neutrophils cultured on uniform substrate or gradient substrate (n≥30 tracks were analyzed for each experiment, N=3 independent experiments for each condition, replicates are biological). All error bars are SEM. ****p<0.0001, ns, not significant, by Student's *t*-test. (**J**) Representative time-lapse images of differentiated HL-60 (dHL-60) cells migrating on fibronectin-coated polyacrylamide gels with uniform stiffness (top) or stiffness gradient (bottom). Color lines indicated the migration trajectories of a single cell. Scale bar: 50 µm. (**K**) Representative tracks of migrating dHL-60 cells on the uniform stiffness gels (top) or stiffness gradient gels (bottom, with greater stiffness towards the top). (**L**) Angular displacement of dHL-60 cells on the uniform stiffness gels (top) or stiffness gradient gels (bottom, with greater stiffness towards the top). (**M**) y-FMI, velocity, and migration persistence of dHL-60 cells cultured on uniform substrate or gradient substrate (n≥30 tracks were analyzed for each experiment, N=3 independent experiments for each condition, replicates are biological). All error bars are SEM. ****p<0.0001, ns, not significant, by Student's *t*-test.

The online version of this article includes the following figure supplement(s) for figure 1:

**Figure supplement 1.** Stiffness gradient of PA-gels, distribution of Paxillin in T cells and MDA-MB-231 cells, and trajectories of migrating neutrophils on gradient gel.

**Figure supplement 2.** Representative trajectories of three different conditions of persistence value when the yFMI1 >yFMI2.

**Figure supplement 3.** Distribution of Paxillin in dHL-60 cells.

(PA) gels (*Figure 1A* iv) with stiffness gradient (20.77 kPa/mm, *Figure 1—figure supplement 1A*) or uniform stiffness (*Rong et al., 2021*). Cell migration trajectories from time-lapse imaging were then generated (*Figure 1A* v) and analyzed for directionality (*Figure 1A* vi). We employed the forward migration index (FMI), for which the displacement value was divided by the total displacement the cell underwent (*Figure 1A* vi) (*DuChez et al., 2019*; *Wu et al., 2012*; *Rong et al., 2021*; *Wang et al., 2021*). This FMI is decoupled with cell velocity and differs from persistence (*Figure 1—figure supplement 2*).

We then performed time-lapse imaging to record CD4+ Naïve T cell motility and generated trajectories for direction analysis. As reported previously (*Shannon et al., 2019*), we observed that T cells underwent typical amoeboid migration with relatively high speed (the mean velocity was 16.21 µm/min) (*Figure 1B and E*, *Videos 1 and 2*), distinguished from mesenchymal migration which is usually below 1 µm/min (*Bear and Haugh, 2014*). When we stained these T cells for focal adhesion marker paxillin (*Wang et al., 2021*), we detected diffused signals by ultrastructure expansion microscopy (U-ExM) (*Figure 1—figure supplement 1B*), distinguished from the plaque-like structure in mesenchymal cells such as the MDA-MB-231 (*Figure 1—figure supplement 1C*).

The trajectories of T cells on the uniform gel did not show a biased direction (*Figure 1B–D*, *Video 1*), while in striking contrast, a majority of cells on the gradient gel exhibited movement towards the stiff-end (*Figure 1B–D*, *Video 2*). FMI for cells on the uniform gel was around zero (indicating random motility) (*Figure 1E*), while it displayed a positive value for cells on the stiffness gradient gel and significantly exceeded that of

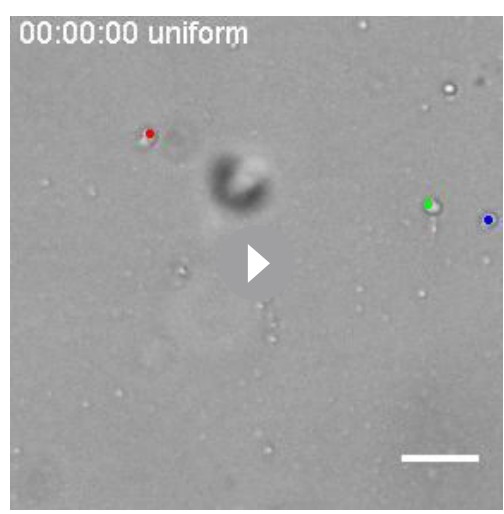

**Video 1.** Representative video of fast crawling CD4+ Naïve T cells on uniform gel in *Figure 1B*. Color lines indicated the migration trajectories of a single cell. Scale bar: 50 µm.

https://elifesciences.org/articles/96821/figures#video1

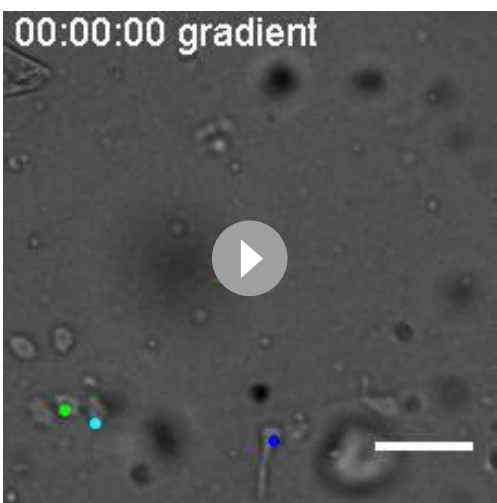

**Video 2.** Representative video of fast crawling CD4⁺ Naïve T cells on gradient gel in *Figure 1B*. Color lines indicated the migration trajectories of single cell. Scale bar: 50 μm.

https://elifesciences.org/articles/96821/figures#video2

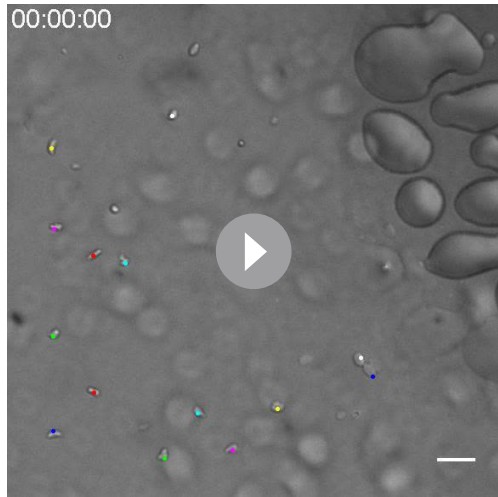

**Video 4.** Representative video of fast crawling neutrophils on gradient gel in *Figure 1—figure supplement 1D*. Color lines and numbers indicated the migration trajectories of a single cell. Scale bar: 50 μm.

https://elifesciences.org/articles/96821/figures#video4

cells on the uniform gel (*Figure 1E*). Angular displacement comparison also revealed that T cells migrated towards the stiff end in this experimental setting (*Figure 1D*). Besides, no obvious difference in velocity was detected when comparing T cells moving on uniform vs gradient gel (*Figure 1E*), indicating that stiffness gradient did not enhance motility but rather increased forward migratory tendency.

Having identified durotaxis in T cells, we next sought to evaluate in other amoeboid cells such as neutrophils whether they also possess the durotactic capacity. Bone marrow-derived neutrophils were isolated by Percoll gradient and confined between two stiffness gradient gels. In line with our observations with T cells, neutrophils displayed random gliding on the uniform gel while appeared to durotax on the gradient gel, as revealed by the FMI and angular displacement analysis (*Figure 1F–I*, *Video 3*). This prominent stiff-end directed movement may result from *Trepat et al., 2012* initial cell polarization upon seeding, exemplified by subsequent persistent migration towards the cell front; or (*Kindberg et al., 2020*) cells frequently exploring and re-steering to the stiffer side of the substrate. When analyzing live cell trajectories, we found that some cells indeed exhibited highly persistent migration (*Figure 1—figure supplement 1D*, Track 1, 7, 9, 14, *Video 4*). However, we also detected cells that frequently changed directions but showed stiff-end biased migration on the whole (*Figure 1—figure supplement 1D*, Track 4, 5, 8, 10, *Video 4*). These observations indicated that neutrophils were able to sense stiffness differences and re-polarize to durotax. Analysis of the bulk cell population revealed that neutrophils on stiffness gradient gel did not elevate their persistence compared to those on the uniform gel (*Figure 1I*). We also employed an immortalized promyelocytic leukemia cell line HL-60 in addition to the mouse-derived neutrophils. In agreement with previous reports, we detected that HL-60

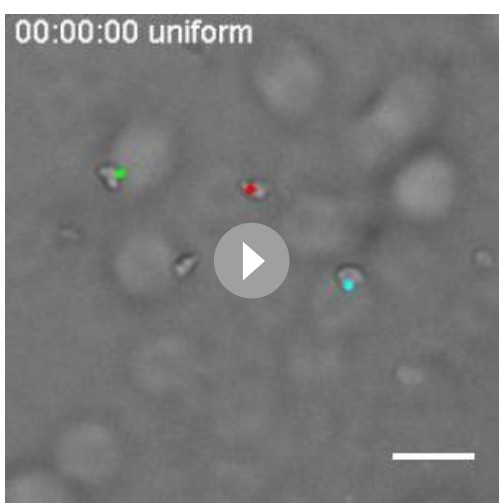

**Video 3.** Representative video of fast crawling neutrophils on both uniform (first 60 frames) and gradient gel (last 60 frames) in *Figure 1F*. Color lines indicated the migration trajectories of a single cell. Scale bar: 50 μm.

https://elifesciences.org/articles/96821/figures#video3

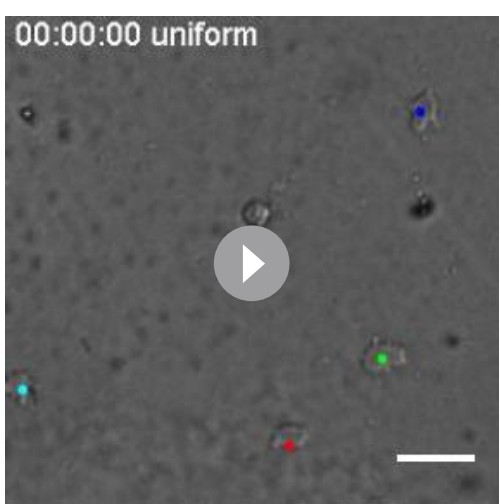

**Video 5.** Representative video of fast crawling dHL-60 cells on both uniform (first 60 frames) and gradient gel (last 60 frames) in *Figure 1J*. Color lines indicated the migration trajectories of a single cell. Scale bar: 50 μm. https://elifesciences.org/articles/96821/figures#video5

employs amoeboid migration when differentiated into neutrophil-like cells (dHL-60) (*Collins et al., 1978*; *Figure 1J*, *Figure 1—figure supplement 3*, *Video 5*). dHL-60 cells migrated randomly on the uniform gel while moving towards the stiff end on gradient gel (*Figure 1J–M*, *Video 5*) under confinement.

Together, these phenomena indicate that amoeboid cells such as T cells and neutrophils, although lacking stable focal adhesions, are capable of sensing the stiffness gradient and orchestrate directional migration towards the stiffer side.

## Amoeboid durotaxis may not be propelled by differential actin flow

Having observed that amoeboid cells can exert durotactic migration, we then interrogated the molecular underpinnings. Classic molecular clutch model attributed both single and collective mesenchymal durotaxis to the differential adhesion strength on distinct substrate stiffness (*Sunyer et al., 2016*; *Plotnikov et al., 2012*; *Bangasser et al., 2017*). Enhanced traction strain energy on the stiffer matrix (*Bangasser et al., 2017*; *Shin et al., 2010*) indicates the presence of increased motors and clutches, which may then slow down the actin flow at adhesion sites by friction (*Bangasser et al., 2017*). In this sense, actin flow is mechanosensitive – with low speed on stiff substrate and high speed on soft matrix. To test the rigidity sensitivity of actin flow in amoeboid cells, we performed quantitative fluorescent speckle microscopy (qFSM) to monitor fluorescent particle movement in dHL-60 cells stained with the actin-specific dye SiR-Actin. Retrograde flow analysis (*Mendoza et al., 2012*) revealed that actin flow velocity dropped sharply when cells were plated on the stiffer substrate (160 kPa) compared to the soft gel (4 kPa) (*Figure 2A*, *Videos 6 and 7*). Moreover, a descending trend of actin flow speed was detected with increasing substrate stiffness (*Figure 2A*, *Videos 6–10*), suggesting that actin flow in amoeboid cells can indeed respond to stiffness.

Actin flow is generated by monomer assembly at the leading edge and myosin contraction at the cell rear, while is slowed down by traction stress between the cell bottom and the substrate (*Gardel et al., 2008*). Strain energy from the halting of actin flow then transduces into a protruding force to counteract membrane tension at the cell front (*Hons et al., 2018*; *Lim et al., 2010*). Thus, the protrusion rate is inversely related to the actin retrograde flow (*Jurado et al., 2005*). It has been suggested that continuous cycles of protrusion support forward cell migration (*Yolland et al., 2019*). Considering the negatively correlated actin flow speed and substrate stiffness (*Figure 2A*), we postulated that differential actin flow velocity (high speed in the down-gradient vs low speed in the up-gradient region) may promote stiff end-biased cell migration by inducing more protrusive events in the up-gradient cell region. To test this, we divided the cell body area into up- and down-gradient halves for respective actin flow analysis. However, against our hypothesis, actin flow in these two regions appeared similar in speed (*Figure 2B*, *Video 11*). To further explore the role of actin flow in durotaxis, we took advantage of CK-666, an Arp2/3 complex inhibitor that prevents branched actin formation, to decrease actin flow speed without changing NMII localization (*Yang et al., 2012*; *Henson et al., 2015*). Upon CK-666 treatment, we observed a significant decrease in actin flow speed (*Figure 2C*) while the durotactic behavior was not impacted (*Figure 2D*), indicating that actin flow may not play a crucial role in amoeboid durotaxis.

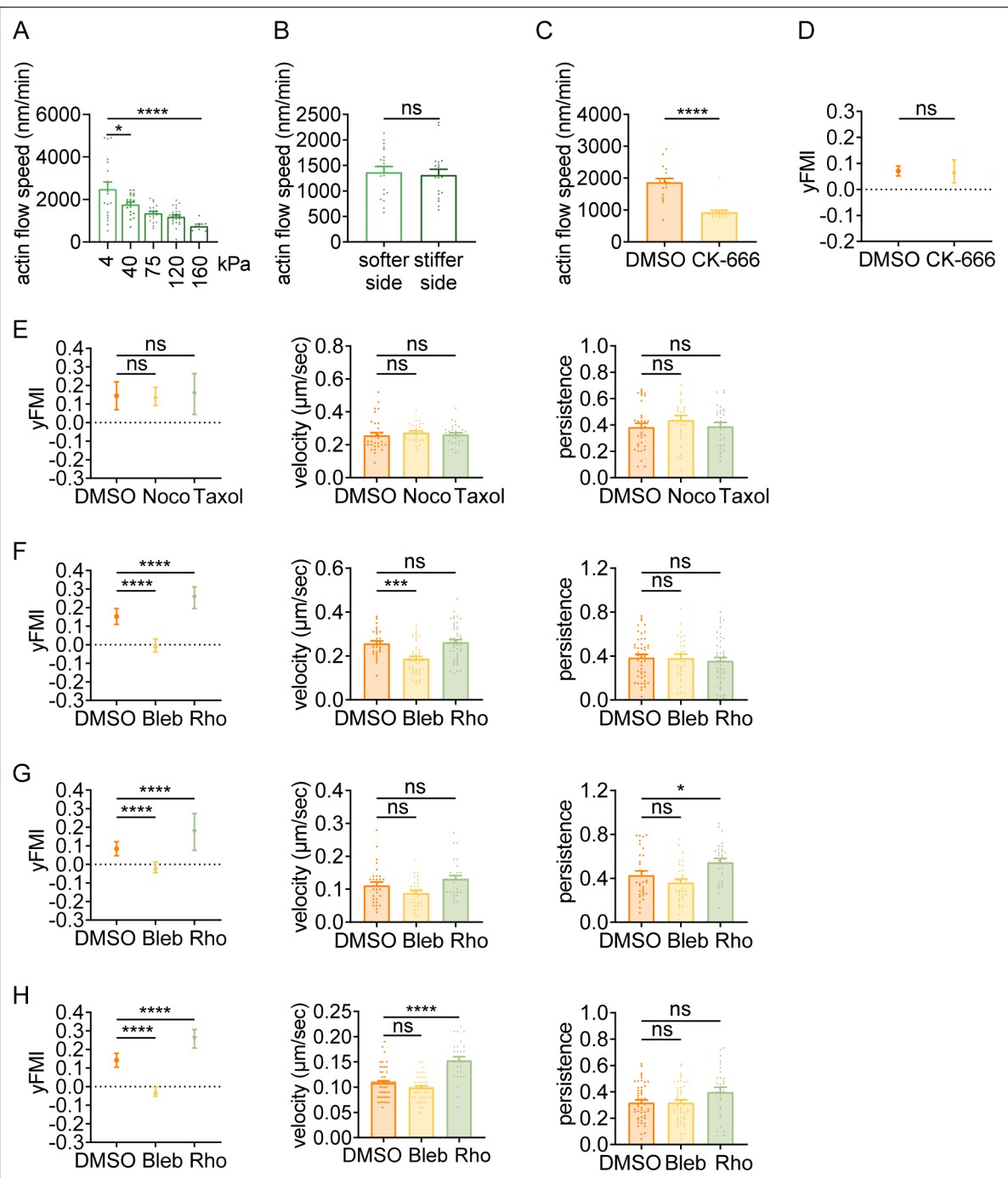

**Figure 2.** Amoeboid durotaxis may not be propelled by differential actin flow. (**A**) Actin flow speed of dHL-60 cells adhered to different substrate stiffness (4 kPa, 40 kPa, 75 kPa, 120 kPa, and 160 kPa). Each plot indicated the average actin flow speed of every individual cell adhered to a polyacrylamide gel. All error bars are SEM. *p<0.05, ****p<0.0001, by one-way ANOVA. (**B**) Actin flow speed of the softer side and stiffer side of the dHL-60 cells adhered on gradient substrate. Each plot indicated the average actin flow speed of the softer or stiffer part of a cell adhered on stiffness gradient gel. All error bars are SEM. ns, not significant, by Student's *t*-test. (**C**) Actin flow speed of control (DMSO) and CK-666 (100 μM, pre-treated for 5 hr) treated CD4[+] Naïve T cells moving on stiffness gradient gel (n=20 tracks were analyzed for each experiment, N=3 independent experiments for each condition, replicates are biological). All error bars are SEM. ****p<0.0001, by Student's *t*-test. (**D**) y-FMI of control (DMSO) and CK-666 (100 μM, pre-treated for 5 hr) treated CD4[+] Naïve T cells moving on stiffness gradient gel (n≥30 tracks were analyzed for each experiment, N=3 independent experiments for each condition, replicates are biological). All error bars are SEM. ns, not significant, by Student's *t*-test. (**E**) y-FMI, velocity, and migration persistence of control (DMSO), nocodazole (Noco, 32 μM, pre-treated for 10 min) treated and taxol (70 nM, pre-treated for 10 min) treated CD4[+] Naïve T cells moving on stiffness gradient gel (n≥30 tracks were analyzed for each experiment, N=3 independent experiments for each condition, replicates are

*Figure 2 continued on next page*

*Figure 2 continued*

biological). All error bars are SEM. ns, not significant, by one-way ANOVA. (**F**) y-FMI, velocity and migration persistence of control (DMSO), blebbistatin (Bleb, 10 µM, pre-treated for 10 min) treated and Rho activator II (Rho, 0.25 µg/mL, pre-treated for 2 hr) treated CD4+ Naïve T cells moving on stiffness gradient gel (n≥30 tracks were analyzed for each experiment, N=3 independent experiments for each condition, replicates are biological). All error bars are SEM. ***p<0.001, ****p<0.0001, ns, not significant, by one-way ANOVA. All drugs were incubated for the whole experiment. (**G**) y-FMI, velocity and migration persistence of control (DMSO), blebbistatin (Bleb, 10 µM, pre-treated for 10 min) treated and Rho activator II (Rho, 0.25 µg/mL, pre-treated for 2 hr) treated neutrophils moving on stiffness gradient gel (n≥30 tracks were analyzed for each experiment, N=3 independent experiments for each condition, replicates are biological). All error bars are SEM. *p<0.05, ****p<0.0001, ns, not significant, by one-way ANOVA. All drugs were incubated for the whole experiment. (**H**) y-FMI, velocity and migration persistence of control (DMSO), blebbistatin (Bleb, 10 µM, pre-treated for 10 min) treated and Rho activator II (Rho, 0.25 µg/mL, pre-treated for 2 hr) treated dHL-60 cells moving on stiffness gradient gel (n≥30 tracks were analyzed for each experiment, N=3 independent experiments for each condition, replicates are biological). All error bars are SEM. ****p<0.0001, ns, not significant by one-way ANOVA.

The online version of this article includes the following figure supplement(s) for figure 2:

**Figure supplement 1.** Effect of Nocodazole and Taxol treatment on microtubule network.

## Durotaxis of T cells and neutrophils is regulated by polarized actomyosin activity

Current mechanistic models explaining directional cell migration along or against a stiffness gradient such as focal adhesion tugging are not plausible in the setting of amoeboid cells. The microtubule dynamics have been reported to impact amoeboid migration (*Kopf et al., 2020*). Besides, we previously found that perturbation of the microtubule network - especially loss of the Golgi microtubules - abrogated single-cell durotaxis in HRPE cells (*Rong et al., 2021*). However, no obvious alterations in durotactic FMI were detected under treatments of either the microtubule polymerization inhibitor Nocodazole or the depolymerization inhibitor Taxol (*Figure 2E*, *Figure 2—figure supplement 1*, *Videos 12 and 13*; *Rong et al., 2021*; *Tabdanov et al., 2021*), indicating that similar influence of the microtubule network on mesenchymal cells may not apply to amoeboid durotaxis.

Recently, the cell rear-localized actomyosin network has been inferred to determine polarity re-establishment by mediating contractility underneath the plasma membrane and propels the locomotion of the cell body. We then asked whether this machinery could provide a steering force for focal adhesion-independent durotaxis. Interestingly, treating T cells or neutrophils with the NMII inhibitor blebbistatin to disrupt cell contraction drastically reduced durotactic FMI towards zero (*Figure 2F and G*, *Video 14*). In contrast, Rho activator II, which can enhance myosin contractility, elevated the FMI of durotactic T cells and neutrophils (*Figure 2F and G*), supporting the notion that NMII activation may play an important role in amoeboid durotaxis. Pharmacological perturbations in dHL-60 cells with blebbistatin or Rho activator II showed similar effects to what we observed in T cells (*Figure 2F and H*), agreeing with myosin activity being pivotal for amoeboid durotaxis.

It has been documented that NMIIB is more sensitive to stiffness gradient than NMIIA in mesenchymal stem cells (*Raab et al., 2012*). Having observed that NMII inhibition significantly decreased FMI in T cells and neutrophils, we next set to explore the roles of specific NMII isoforms in amoeboid durotaxis. We generated HL-60 cells knocked down of NMIIA or NMIIB (*Figure 3A*, *Figure 3—figure supplement 1A*). Interestingly, cell trajectories on gradient gel revealed that loss of NMIIA rather than

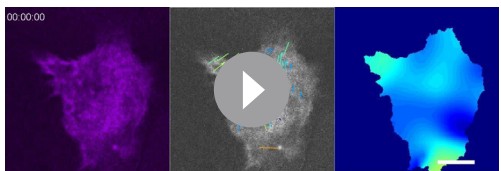

**Video 6.** Representative video of dHL-60 cell stained with 0.1 µM SiR-Actin on 160 kPa substrate. Left: live cell video of dHL-60 cell. Middle: raw video of left panel integrated with actin flow vectors. Right: heat map of actin flow distribution for the left panel. The video is captured at 15- s intervals for 10 min. Scale bar: 10 µm.
https://elifesciences.org/articles/96821/figures#video6

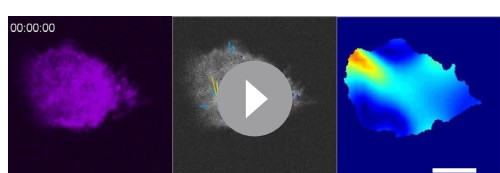

**Video 7.** Representative video of dHL-60 cell stained with 0.1 µM SiR-Actin on 4 kPa substrate. Left: live cell video of dHL-60 cell. Middle: raw video of left panel integrated with actin flow vectors. Right: heat map of actin flow distribution for left panel. The video is captured at 15- s intervals for 10 min. Scale bar: 10 µm.
https://elifesciences.org/articles/96821/figures#video7

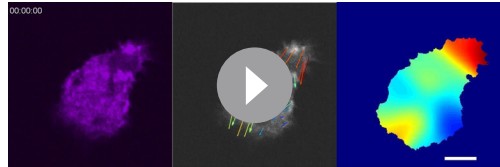

**Video 8.** Representative video of dHL-60 cell stained with 0.1 µM SiR-Actin on 40 kPa substrate. Left: live cell video of dHL-60 cell. Middle: raw video of left panel integrated with actin flow vectors. Right: heat map of actin flow distribution for the left panel. The video is captured at 15- s intervals for 10 min. Scale bar: 10 µm.
https://elifesciences.org/articles/96821/figures#video8

**Video 10.** Representative video of dHL-60 cell stained with 0.1 µM SiR-Actin on 120 kPa substrate. Left: live cell video of dHL-60 cell. Middle: raw video of left panel integrated with actin flow vectors. Right: heat map of actin flow distribution for the left panel. The video is captured at 15- s intervals for 10 min. Scale bar: 10 µm.
https://elifesciences.org/articles/96821/figures#video10

NMIIB abolished dHL-60 durotaxis (*Figure 3B–E*, *Video 15*), which was not in accordance with what has been reported in mesenchymal cells (*Raab et al., 2012*). To exclude functional compensation between these two NMII isoforms, we performed a Western blot to detect protein levels of these two isoforms in NMIIA[KD] and NMIIB[KD] cells. The results showed that NMIIB protein level in NMIIA[KD] cells doubled while there was no compensatory upregulation of NMIIA in NMIIB[KD] cells (*Figure 3—figure supplement 1B, C*), further suggesting NMIIA rather than NMIIB is responsible for amoeboid durotaxis since in NMIIA[KD] cells, compensatory upregulation of NMIIB did not rescue the durotaxis-deficient phenotype.

Previous studies showed that when mesenchymal cells crawl on rigid glass, NMIIB is enriched at the cell rear (*Sandquist et al., 2006*), whereas NMIIA appears uniformly distributed (*Raab et al., 2012*). To examine the mechanism underlying the requirement for NMIIA but not NMIIB in amoeboid durotaxis, we assayed their distribution in cells on gradient gel. Strikingly, NMIIA was concentrated at the down-gradient side of dHL-60 cells, regardless of the front-rear axis of the cell (*Figure 3F and G*). On the contrary, NMIIB showed much less correlation with the stiffness gradient (*Figure 3G*), consistent with it playing a minor role in the establishment of durotactic direction. Similar distribution of NMIIA was also seen in T cells and neutrophils on gradient gel (*Figure 3H*, *Figure 3—figure supplement 1D, E*).

Previous studies suggested that localizing NMIIA to the cell rear is required to set cell polarity (*Raab et al., 2012*; *Jacobelli et al., 2004*). We then hypothesized that the down-gradient concentrated NMIIA may leverage the cell front towards the stiffer side, thus enabling efficient amoeboid durotaxis. To test this, we transfected HL-60 cells with GFP-NMIIA and monitored the fluorescent signal in durotaxis assays. We captured events when cells turned to align with the stiffness gradient and in these cases, the GFP-NMIIA signal stayed concentrated at the soft side, preceding re-polarization of the cell to form a new front-rear axis (*Figure 3—figure supplement 2A*, *Video 16*).

## Amoeboid durotaxis is evolutionarily conserved

Having identified durotactic behavior in T cells and neutrophils, we then asked whether this low-adhesive, rapid-moving durotaxis was evolutionarily conserved. To answer this, we switched to the

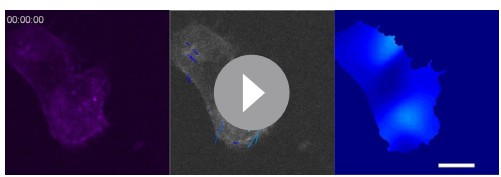

**Video 9.** Representative video of dHL-60 cell stained with 0.1 µM SiR-Actin on 75 kPa substrate. Left: live cell video of dHL-60 cell. Middle: raw video of left panel integrated with actin flow vectors. Right: heat map of actin flow distribution for the left panel. The video is captured at 15- s intervals for 10 min. Scale bar: 10 µm.
https://elifesciences.org/articles/96821/figures#video9

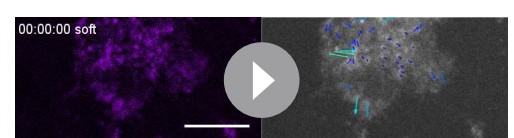

**Video 11.** Representative video of both soft (first 40 frames) and stiff parts (last 40 frames) of dHL-60 cell stained with 0.1 µM SiR-Actin on gradient gel in *Figure 2B*. Left: live cell video of dHL-60 cell. Right: raw video of left panel integrated with actin flow vectors. The video is captured at 2- s intervals for 80 sec. Scale bar: 10 µm.
https://elifesciences.org/articles/96821/figures#video11

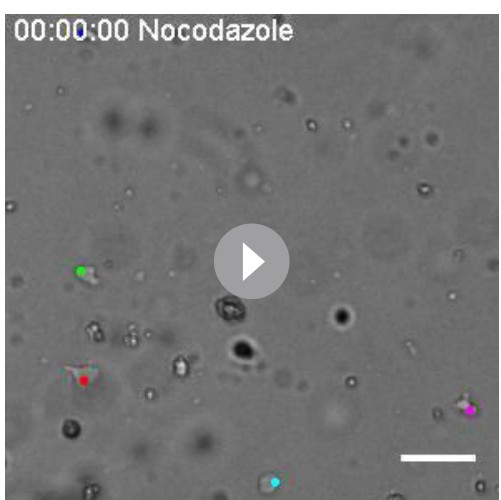

**Video 12.** Representative video of fast crawling CD4⁺ Naïve T cells on gradient gel with 32 µM Nocodazole treatment in *Figure 2E*. Scale bar: 50 µm.

https://elifesciences.org/articles/96821/figures#video12

protista *Dictyostelium*, which moves in amoeboid mode when reside as single cells. After being plated on the PA-gel surface under confinement, *Dictyostelium* exhibited elongated morphology and extended convex membrane protrusions at the front as described before (*Van Haastert, 2011*), which then alternated sideways for gliding (*Figure 4A*, *Video 17*). Agreeing with our previous findings with T cells and neutrophils, *Dictyostelium* also displayed stiff-side directed motility when seeded on the gradient gel (*Figure 4A–D*, *Video 17*). Blebbistatin application drastically dropped the durotactic FMI to around zero (*Figure 4E*), while Rho activator II treatment significantly elevated this value (*Figure 4E*). Moreover, in line with what we discovered in T cells, microtubule perturbation did not hamper the directionality of *Dictyostelium* durotaxis (*Figure 4F*). These results suggest that the molecular mechanism underlying amoeboid durotaxis is largely conserved in *Dictyostelium*.

## Simulation of amoeboid durotaxis with a mechanosensitive active gel model

To better characterize amoeboid durotaxis observed in our experiments, we established an active gel model (*Tjhung et al., 2012*), in which a cell is represented by an active droplet (*Figure 5A*) $c\left(\mathbf{r}, t\right)$ composed of actin filaments with a certain density, whose polarization field is defined by $\mathbf{P}\left(\mathbf{r}, t\right)$, with $\mathbf{r}$ and $t$ denoting the position vector and time, respectively (*Figure 5B*). The free energy $F\left[c, \mathbf{P}\right]$ to confine actin assembly into a droplet can be formulated as (*Tjhung et al., 2012*):

$$F\left[c, \mathbf{P}\right] = \int d^3 r \left\{ \frac{4a}{c_0^4} c^2 \left(c - c_0\right)^2 + \frac{k}{2} |\nabla c|^2 - \frac{\alpha}{2} \frac{\left(2c - c_0\right)}{c_0} |\mathbf{P}|^2 + \frac{\alpha}{4} |\mathbf{P}|^4 + \frac{\kappa}{2} \left(\nabla \mathbf{P}\right)^2 \right\},\qquad (1)$$

This energy ensured that a polar active actin phase $c > c_0/2$ ($|\mathbf{P}| > 0$) and a passive solvent phase $c=0$ ($|\mathbf{P}|=0$) coexist in this system, where $c_0$ denotes

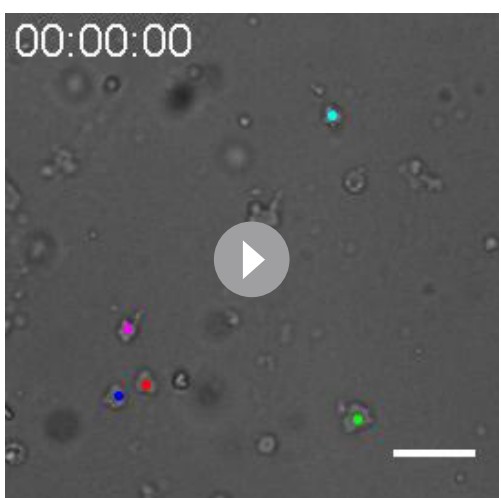

**Video 13.** Representative video of fast crawling CD4⁺ Naïve T cells on gradient gel with 70 nM Taxol treatment in *Figure 2E*. Scale bar: 50 µm.

https://elifesciences.org/articles/96821/figures#video13

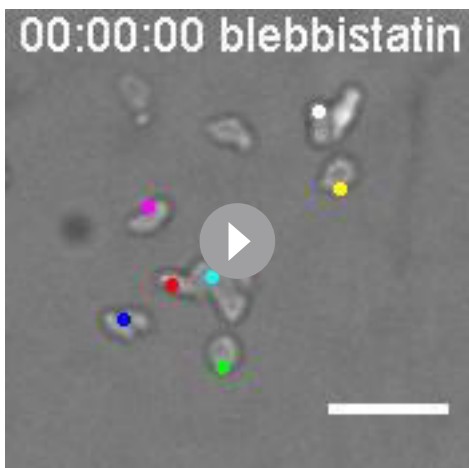

**Video 14.** Representative video of fast crawling neutrophils on gradient gel with 10 µM blebbistatin treatment (first 60 frames) and 0.25 µg/mL Rho activator II treatment (last 60 frames) in *Figure 2G*. Scale bar: 50 µm.

https://elifesciences.org/articles/96821/figures#video14

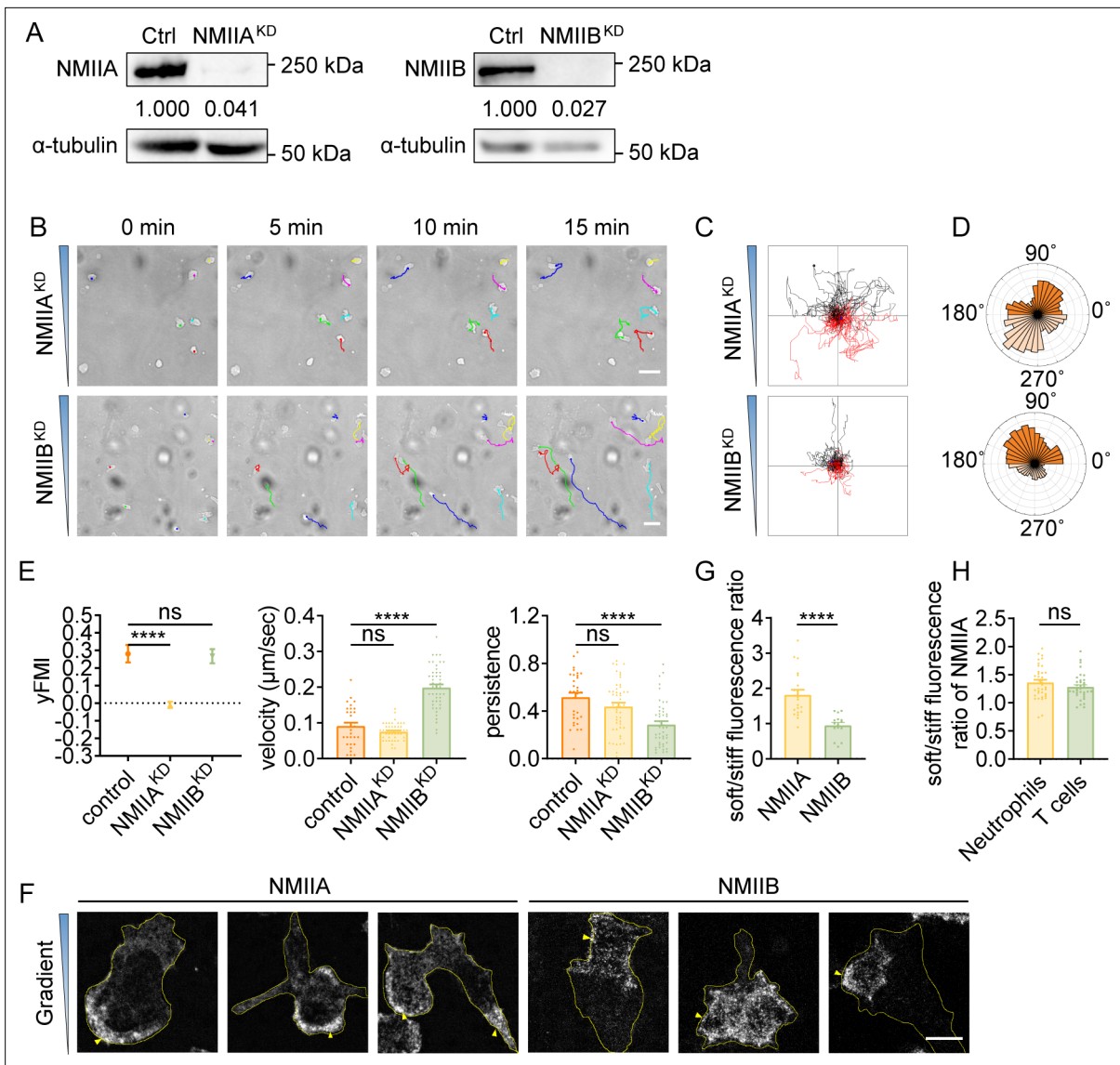

**Figure 3.** Durotaxis of T cells and neutrophils is regulated by polarized actomyosin activity. (**A**) Western blot showing the protein level of non-muscle myosin IIA (NMIIA) (left) and NMIIB (right) in the negative control (Ctrl), NMIIA knocked down (NMIIA^KD) and NMIIB knocked down (NMIIB^KD) HL-60 cells. α-tubulin was used as the loading control. Numbers below the ladders were the relative intensity versus α-tubulin. (**B**) Representative time-lapse images of NMIIA^KD (top) and NMIIB^KD (bottom) dHL-60 cells migrating on polyacrylamide gels with stiffness gradient. Color lines indicated the migration trajectories of a single cell. Scale bar: 50 µm (**C**) Representative tracks of migrating NMIIA^KD (top) and NMIIB^KD (bottom) dHL-60 cells on stiffness gradient gels. (**D**) Angular displacement of NMIIA^KD (top) and NMIIB^KD (bottom) dHL-60 cells on stiffness gradient gels (with greater stiffness towards the top). (**E**) y-FMI, velocity and migration persistence of control, NMIIA^KD and NMIIB^KD dHL-60 cells migrating on stiffness gradient gel (n≥30 tracks were analyzed for each experiment, N=3 independent experiments for each condition, replicates are biological). All error bars are SEM. ****p<0.0001, ns, not significant, by one-way ANOVA. (**F**) Representative immunofluorescent staining of NMIIA (left) and NMIIB (right) of dHL-60 cells on stiffness gradient gel (with greater stiffness towards the top). The yellow triangles indicated the location of NMIIA and NMIIB. Scale bar: 10 µm. (**G**) Fluorescent intensity ratio of NMIIA or NMIIB in cell softer part to stiffer part in F. All error bars are SEM. ****p<0.0001, by Student's *t*-test. (**H**) Fluorescent intensity ration of NMIIA in softer part to stiffer part of neutrophils and CD4+ Naïve T cells in *Figure 3—figure supplement 1D and E*. All error bars are SEM. ns, not significant, by Student's *t*-test.

The online version of this article includes the following source data and figure supplement(s) for figure 3:

**Source data 1.** Original files for Western blot in *Figure 3A*.

**Source data 2.** Western blot with labeled bands in *Figure 3A*.

*Figure 3 continued on next page*

**Figure supplement 1.** Protein level of NMIIA and NMIIB in NMIIA^KD and NMIIB^KD HL-60 cells and distribution of NMIIA in T cells and neutrophils.

**Figure supplement 1—source data 1.** Original files for Western blot in *Figure 3—figure supplement 1B*.

**Figure supplement 1—source data 2.** Western blot with labeled bands in *Figure 3—figure supplement 1B*.

the average actin density. $\alpha$ is the stability constant and $k$ determines the interfacial tension between actin and the adjacent solvent phase. $a$ is a phenomenological free energy amplitude to define the bulk property of the droplet, and $\kappa$ describes the elastic constant of liquid-crystalline elasticity. In addition, fluid velocity $\mathbf{v}$, which is involved in actin convection, is incorporated here. Cell migration is described by the spatial evolution of actin density $c(\mathbf{r}, t)$ and can be influenced by the fluid velocity and polarization field. The evolution of actin density is formulated as (*Tjhung et al., 2012*):

$$\frac{\partial c}{\partial t} + \nabla \cdot \left[ c \left( \mathbf{v} + w\mathbf{P} \right) - M\nabla \frac{\delta F}{\delta c} \right] = 0, \tag{2}$$

where $M$ is the thermodynamic mobility parameter, $w$ denotes the polymerization rate of the actin filament, and $F$, expressed by *Equation 1*, is the free energy that constrains actins within the droplet. The incompressible Navier–Stokes equation is invoked to describe fluid velocity $\mathbf{v}$ (*Tjhung et al., 2012*; *Tjhung et al., 2015*):

$$\nabla \cdot \mathbf{v} = 0,$$
$$\rho \left( \frac{\partial}{\partial t} + \mathbf{v} \cdot \nabla \right) \mathbf{v} = -\nabla P + \nabla \cdot \boldsymbol{\sigma}^{\text{total}} - \gamma \mathbf{v}, \tag{3}$$

where $P$ is the isotropic pressure and $\gamma$ is the friction coefficient. Total hydrodynamic stress $\boldsymbol{\sigma}^{\text{total}}$ includes four parts, including viscous stress $\boldsymbol{\sigma}^{\text{vis}}$ (Materials and methods, Active gel model, *Equation 7*), elastic stress $\boldsymbol{\sigma}^{\text{ela}}$ (Materials and methods, Active gel model, *Equation 8*), interfacial stress between active and passive phase $\boldsymbol{\sigma}^{int}$ (Materials and methods, Active gel model, *Equation 9*), and active stress $\boldsymbol{\sigma}^{\text{act}}$ (*Hatwalne et al., 2004*). The active contraction of cells is represented by the active stress $\boldsymbol{\sigma}^{\text{act}} = \bar{\zeta} c\mathbf{P}\mathbf{P}$, where $\bar{\zeta}$ measures the contractile strength.

NMIIA is related to cell polarity establishment during cell durotaxis (*Raab et al., 2012*). In our amoeboid durotaxis experiments, we observed soft-end polarized NMIIA (*Figure 3—figure*

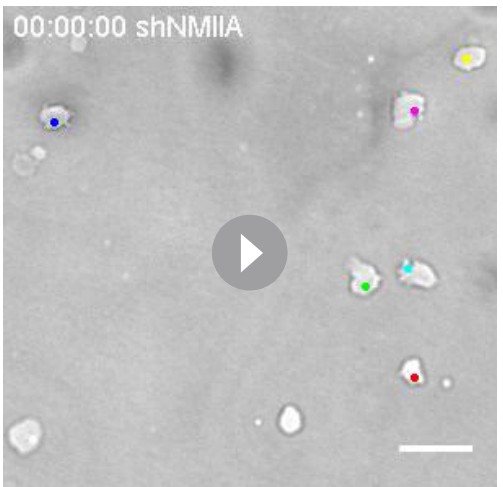

**Video 15.** Representative video of both NMIIA^KD (first 60 frames) and NMIIB^KD dHL-60 cells (last 60 frames) migrating on gradient gel in *Figure 3B*. Color lines indicated the migration trajectories of a single cell. Scale bar: 50 µm.

https://elifesciences.org/articles/96821/figures#video15

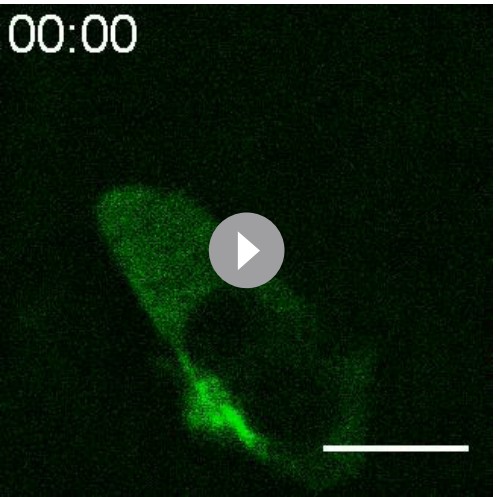

**Video 16.** Representative video of a GFP-NMIIA dHL-60 cell migrating on gradient gel in *Figure 3—figure supplement 2A*. Scale bar: 10 µm.

https://elifesciences.org/articles/96821/figures#video16

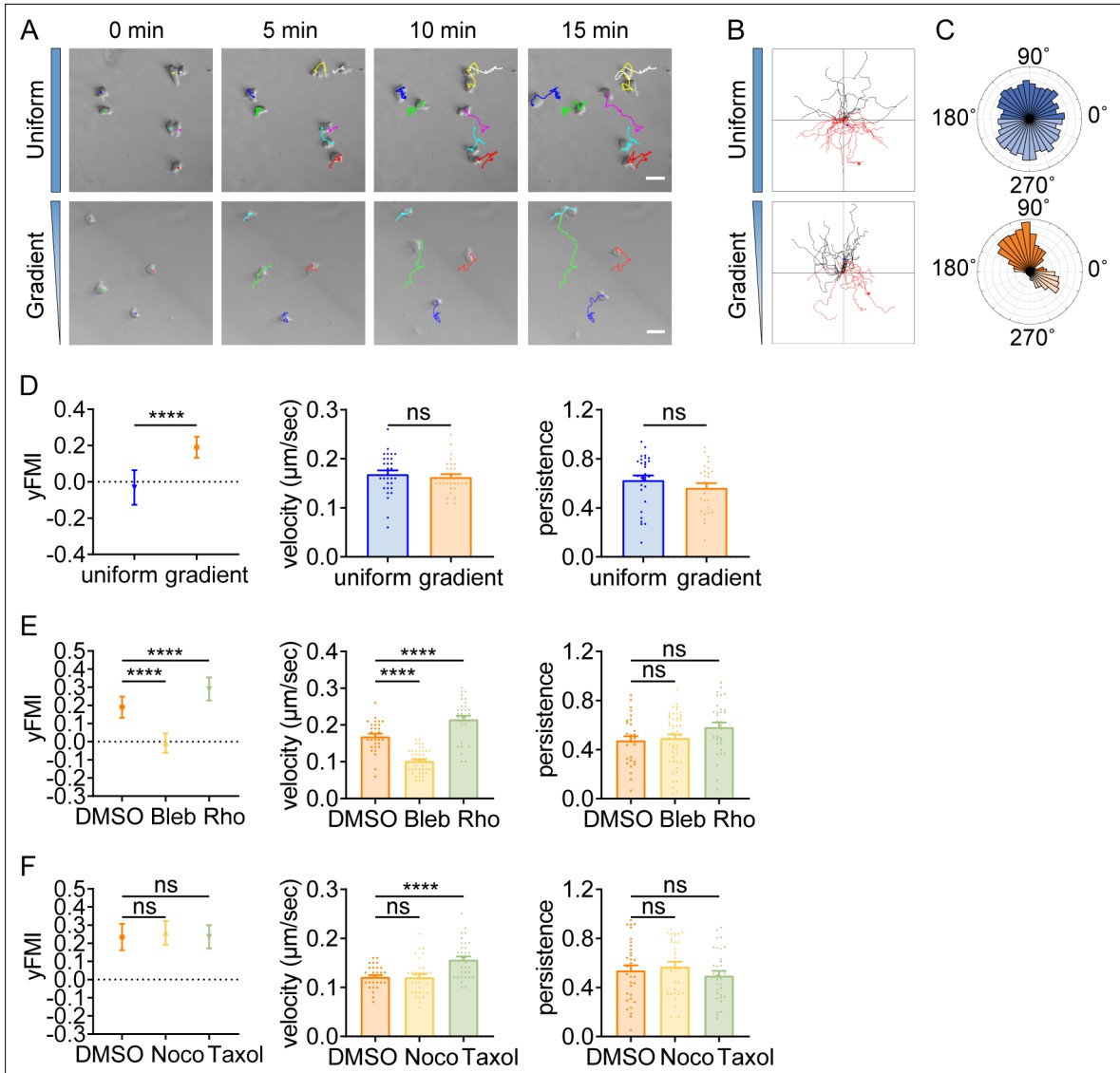

**Figure 4.** Amoeboid durotaxis is evolutionarily conserved. (**A**) Representative time-lapse images of *Dictysotelium* on polyacrylamide gels with uniform stiffness (top) or stiffness gradient (bottom). Color lines indicated the migration trajectories of single cell. Scale bar: 50 µm. (**B**) Representative tracks of migrating *Dictysotelium* on the uniform stiffness gels (top) or stiffness gradient gels (bottom, with greater stiffness towards the top). (**C**) Angular displacement of *Dictysotelium* on the uniform stiffness gels (top) or stiffness gradient gels (bottom, with greater stiffness towards the top). (**D**) y-FMI, velocity and migration persistence for *Dictysotelium* cultured on uniform substrate or gradient substrate (n≥30 tracks were analyzed for each experiment, N=3 independent experiments for each condition, replicates are biological). All error bars are SEM. ****p<0.0001, ns, not significant, by Student's *t*-test. (**E**) y-FMI, velocity, and migration persistence of control (DMSO), blebbistatin (Bleb,10 µM, pre-treated for 10 min) treated and Rho activator II (Rho, 0.25 µg/mL, pre-treated for 2 hr) treated *Dictysotelium* moving on stiffness gradient gel (n≥30 tracks were analyzed for each experiment, N=3 independent experiments for each condition, replicates are biological). All error bars are SEM. ****p<0.0001, ns, not significant, by one-way ANOVA. (**F**) y-FMI, velocity and migration persistence of control (DMSO), nocodazole (Noco, 32 µM, pre-treated for 10 min) treated and taxol (70 nM, pre-treated for 10 min) treated *Dictysotelium* moving on stiffness gradient gel (n≥30 tracks were analyzed for each experiment, N=3 independent experiments for each condition, replicates are biological). All error bars are SEM. ****p<0.0001, ns, not significant, by one-way ANOVA.

*supplement 2A*), which is involved in the present model. Thus, we propose that the density of NMIIA *m* evolves as (*Genkin et al., 2017*):

$$\frac{\partial m}{\partial t} + \nabla \cdot \left[ m \left( \mathbf{v} - w\mathbf{P} \right) \right] = D_m \left( E \right) \nabla \cdot \nabla m - \frac{1}{\Gamma_m} \left[ -\text{sign} \left( 2c - c_0 \right) \alpha m + \alpha \frac{m^3}{m_0^2} \right], \quad (4)$$

where $D_m \left( E \right)$ denotes diffusive ability of NMIIA and is related to the substrate stiffness $E$. The diffusion parameter $D_m \left( E \right)$ outside the cell ($c < c_0/2$) is set to be uniform and defined by the basal diffusion parameter $D_0$. Inside the cell, $D_m \left( E \right)$ is defined by

$$D_m \left( E \right) = \frac{\bar{\zeta} D_0}{\bar{\zeta}_{\text{thr}}} \left( \frac{E - E_{min}}{E_{max} - E_{min}} \right)^2 \left[ 1 - \frac{1}{1 + e^{(E_{\text{m}} - E_{\text{aver}})/E_0}} \right], \quad (5)$$

where $E_{min}$ and $E_{max}$ are the minimum and maximum substrate stiffness locally sensed by the cell. $\bar{\zeta}_{\text{thr}}$ is the threshold of cell contractility and cooperates with the substrate stiffness to characterize the sensitivity of cells to the substrate stiffness gradient. $E_{\text{aver}} = \langle E \rangle_{c \geq c_0/2}$ is the average substrate stiffness sensed by the cell. $E_{\text{m}}$ and $E_0$ are the threshold of stiffness sensitivity and the reference substrate stiffness, respectively, which regulate the most sensitive range of stiffness for cells. With the density of actin and NMIIA defined above, the global polarity vector of the cell is determined by the unit vector $\bar{\mathbf{n}}_{\text{polar}}$ of $\mathbf{n}_{\text{polar}} = \sum_{\mathbf{r}} \mathbf{r} / \sum_{\mathbf{r}} m - \langle \mathbf{r} \rangle_{c \geq c_0/2}$, which points from the center of NMIIA to the cell centroid. As evidenced in our experiments, the local polarity of the actin flow $\mathbf{P}$ is aligned to $\bar{\mathbf{n}}_{\text{polar}}$ (*Videos 6–10*). Therefore, the evolution of actin polarity field can be formulated by *Genkin et al., 2017*; *Cates et al., 2009*:

$$\frac{\partial \mathbf{P}}{\partial t} + \left[ \left( \mathbf{v} + w\mathbf{P} \right) \cdot \nabla \right] \mathbf{P} = -\underline{\underline{\Omega}} \cdot \mathbf{P} + \xi \underline{\underline{v}} \cdot \mathbf{P} - \frac{1}{\Gamma} \frac{\delta F}{\delta \mathbf{P}} - \bar{\zeta} \left[ \mathbf{P} \cdot \left( \mathbf{n}_{\text{polar}} \mathbf{R}_{\pi/2} \right) \right] \mathbf{P} \mathbf{R}_{\pi/2}, \quad (6)$$

where $\underline{\underline{\Omega}}$ and $\underline{v}$ are the anti-symmetric and asymmetric part of $\nabla \mathbf{v}$, respectively; $\Gamma$ denotes the rotational viscosity and $\mathbf{R}_{\pi/2}$ is a rotational matrix.

The proposed model successfully recapitulates amoeboid durotaxis found in our experiments (*Figure 5C–E*). Initially, actin polarity $\mathbf{P}$ is assumed to be random (*Figure 5C*). With increasing time, $\mathbf{P}$ tends to align along the stiffness gradient (*Figure 5D*, *Video 18*). In addition, asymmetric distribution of NMIIA is observed in our simulation, where NMIIA displays a high concentration at the rear of the cell (*Figure 5E*), consistent with our experiments (*Figure 3F*).

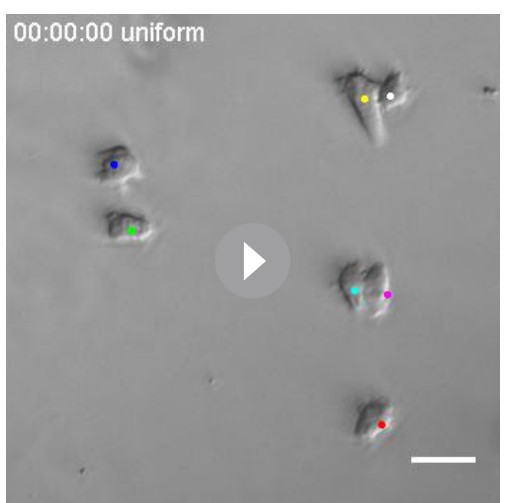

**Video 17.** Representative video of fast crawling *Dictyostelium* cells on both uniform (first 60 frames) and gradient gel (last 60 frames) in *Figure 4A*. Color lines indicated the migration trajectories of a single cell. Scale bar: 50 µm.
https://elifesciences.org/articles/96821/figures#video17

## Cell contractility and substrate stiffness concertedly regulate amoeboid durotaxis

Our experiments showed that decreasing cell contractility can inhibit amoeboid durotaxis. We theoretically demonstrate that with low cell contractility $\bar{\zeta}$, amoeboid durotaxis disappears and random trajectories of cells are observed (*Figure 6A*). With increasing $\bar{\zeta}$, cells indeed perform directional migration along the stiffness gradient (*Figure 6B*). To quantify amoeboid durotaxis in our model, we also calculate the FMI (*Figure 6E*), similar to experimental statistics. Increased FMI indicates stronger amoeboid durotaxis. It is evident that with increasing cell contractility $\bar{\zeta}$, FMI elevates, accompanied by more robust amoeboid durotaxis.

Our model suggests that the basal diffusion rate of NMIIA is larger on the stiffer substrate, improving the mechanosensity of cells to substrate gradient. In addition, the model predicts that cells initially placed on softer regions of gradient

substrates exhibit stronger durotaxis (*Figure 6C–E*). Furthermore, the initial local substrate stiffness may reverse cell durotaxis to a durotaxis. For example, under cell contractility $\bar{\zeta}=0.002$, cells on the softer region of gradient substrate, i.e., initial stiffness of cell centroid $E$=30 and 70 kPa, migrate directionally following the stiffness gradient (*Figure 6C*), whereas the strong durotaxis gives way to random migration when the initial stiffness of cell centroid is 200 kPa (*Figure 6D*), resulting in an FMI close to zero (*Figure 6E*). To test our theoretical predictions, we segregated the gradient gel (ranging from 30 kPa to 210 kPa) into three equal regions (soft, middle, and stiff) (*Figure 6F*) for cell trajectory analysis. FMI of cells on the soft region topped, while cells in the stiff region displayed more random motility (*Figure 6G*). Cell velocity and persistence showed no difference among these regions (*Figure 6H, I*). These experiments agreed with our theoretical predictions, thereby confirming the robustness regulation of amoeboid durotaxis by local stiffness.

## Discussion

Durotaxis based on focal adhesion-dependent mechanosensing has been well studied for decades. However, whether cells can durotax using amoeboid migration strategies was asked only recently (*Espina et al., 2022*). In this study, we developed an imaging-based cell migration device that provided both spatial confinement and stiffness gradient, and with this device, we demonstrated that amoeboid cells can durotaxis. We further uncovered the underlying mechanism to involve NMIIA polarization towards the soft-matrix-side but may not require differential actin flow up- or down-stiffness gradient.

An intriguing question remains is how cells manage to retain NMIIA on regions that downward the stiffness gradient. One likely explanation would be that NMIIA could sense substrate stiffness (*Raab et al., 2012*) and then locate itself on the soft end. In our modeling, we assigned NMIIA with an initial diffusion rate which increases with higher substrate stiffness. Under this consideration, NMIIA is more diffusive on the stiff end and intended to accumulate on the soft end. To experimentally verify this, future work using single particle tracking is required to analyze the NMIIA diffusion rate on the substrate with different stiffness. Another possible mechanism may be that NMIIA is recruited by its upstream factors which prefer to be located in softer areas. Mst1 kinase is a key component of the Hippo pathway that could be regulated by substrate stiffness. It has been reported that Mst1 directly regulates NMIIA dynamics to organize high and low-affinity LFA-1 to the anterior and posterior membrane during T cell migration (*Xu et al., 2014*). Whether Mst1 is mechanosensitive and has the capacity to recruit NMIIA to the soft end still needs to be testified. The mDia1-like formin A, IQGAP-related proteins, and actin-bundling protein cortexillin localize like NMII to the rear of polarized, migrating *Dictyostelium* cells, promoting the formation of cortical actin in the rear and allowing efficient cell migration in 2D-confined environments (*Ramalingam et al., 2015*). Whether these proteins localize to the soft end of durotactic amoeboid cells and recruit NMIIA is an intriguing question. We have used formin inhibitor SMIFH2 in our durotaxis assay and found this treatment slightly decreased migration velocity but did not impact the durotactic behavior of T cells (*Figure 3—figure supplement 2B*). The possibility that IQGAP-related proteins and cortexillin act as upstream factors that recruit NMIIA to the softer area is still open.

Aside from NMIIA, other mechanisms may also facilitate amoeboid durotaxis. Inorganic droplets can move directionally on surfaces with a stiffness gradient due to the intrinsic wettability (*Bueno et al., 2018*; *Pallarès et al., 2023*). Droplets with smaller and wetting contact angles move towards the soft end, while those with larger and non-wetting contact angles can durotax (*Bueno et al., 2018*). We also measured the contact angles of dHL-60 cells at both the stiffer and softer sides on gradient gel and observed non-wetting contact angles (greater than 90 degrees) on both sides (*Figure 3—figure supplement 2C*) albeit the angles on the softer side are larger (*Figure 3—figure supplement 2D*). Hence, it can be postulated that the smaller, non-wetting contact angle on the stiffer side is also likely to contribute to amoeboid durotaxis to a certain extent.

Immune cell activation is regulated not only by biochemical factors, but also by the mechano-microenvironment including substrate stiffness (*Pageon et al., 2018*). Substrate rigidity influences differentiation (*O'Connor et al., 2012*), gene expression, cell migration, morphology, and cytokine secretion (*Saitakis et al., 2017*) in T cells. Interestingly, cytotoxic T lymphocytes (CTL) have been shown to enhance killing response when encountering stiffer target cells (*Basu et al., 2016*). Whether stiff tumor cells can act as a mechanical guidance for CTL durotaxis is tentative to test. Immune cells are

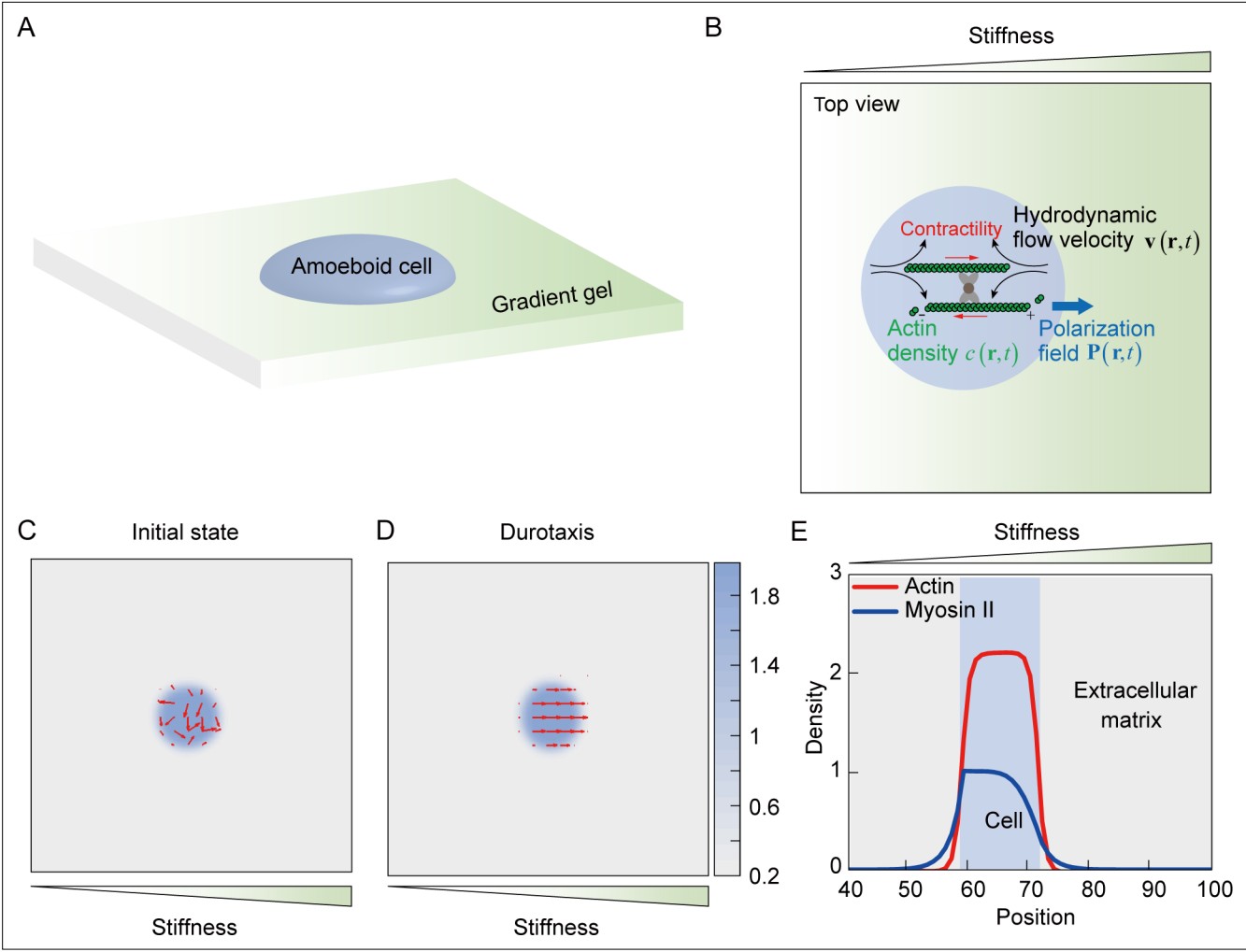

**Figure 5.** Mechanosensing active gel model of amoeboid durotaxis. (**A**) Illustration of an amoeboid cell crawling on the polyacrylamide (PA) gel with stiffness gradient. (**B**) Details of the mechanosensing active gel model. The blue circle represented an amoeboid cell. The position of the blue circle was defined by actin density (green) with a given polarization field. To recapitulate cell migration, myosin contractility (red), and hydrodynamic flow velocity (black) were illustrated in the model. (**C**) Initial density and polarization field of actin. The background color indicated the density of actin both inside and outside of the cell border. The red arrows indicated the polarization field of actin. (**D**) Density and polarization field of actin during amoeboid durotaxis. The background color indicated the density of actin both inside and outside of the cell border. The red arrows indicated the polarization field of actin. (**E**) Spatial distribution of NMIIA (blue line) and actin (red line) along the direction of stiffness gradient. The cell position was marked with a blue rectangle.

recruited and play a crucial role in tissue repair. Tissue injuries of varying origins, including chemical, mechanical, or microbiological, initiate the processes that ultimately result in fibrosis (*Tschumperlin et al., 2018*). Fibrosis is characterized by tissue stiffening due to excessive extracellular matrix protein deposition and cross-linking (*Wells, 2013*). It will be intriguing to investigate whether the rigidity cues in the fibrotic area may regulate immune cell migration.

The search for molecular mechanisms underlying cell durotaxis has been paralleled by the development of different types of theoretical models (*Bangasser et al., 2017*). The well-established motor-clutch model (*Isomursu et al., 2022*) and random walk model (*Doering et al., 2018*) demonstrate that cell durotaxis is attributable to traction or cell-substrate adhesion difference. Vertex model (*Alt et al., 2017*) and continuum model (*Pallarès et al., 2023*) are widely used in collective durotaxis, focusing on long-range environmental stiffness difference and force generation between adjacent cells and suggesting far more efficiency in collective durotaxis than single-cell durotaxis. The active nematic model employs active extensile or contractile agents to push or pull the fluid along their elongation axis to simulate cell flow (*Doostmohammadi et al., 2018*). These models, which usually

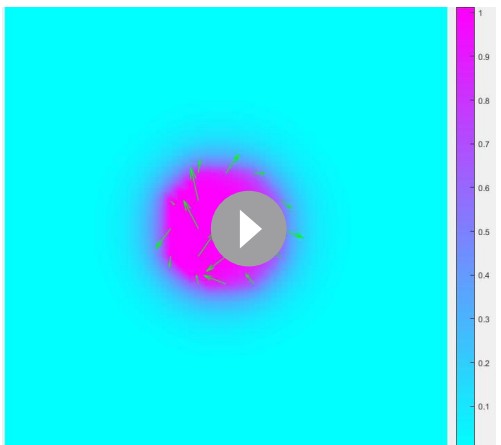

**Video 18.** Representative video of amoeboid durotaxis active gel model. The green arrows indicated the polarization field of actin. The background color indicated the density of myosin both inside and outside of the cell border. The white track indicated the migration trajectory of migrating amoeboid cells.
https://elifesciences.org/articles/96821/figures#video18

ignore the asymmetric distribution of polarity markers, are more suitable for mesenchymal cells forming obvious stress fibers, but might not be applicative in low adhesive amoeboid cells. The mechanosensing active gel model established here highlights the role of polarity markers in the migration of cells with a low adhesion. It also provides us a novel perspective that the diffusion rate difference dependent on substrate stiffness can polarize NMIIA and subsequently induce polarized actin flow along the stiffness gradient, followed by amoeboid durotaxis.

Our study also has technical and mechanistic limitations. Cells live in complex, three-dimensional physiological or pathological microenvironments. Thus, the phenomena observed in our 2D confinement assay may not faithfully recapitulate the behavior of amoeboid cells in vivo. Further measurement of in vivo stiffness gradient sensed by immune cells and more in vivo studies can broaden our understanding of amoeboid durotaxis. In addition, mechanistic studies of amoeboid durotaxis in our research is limited. We found that soft end polarized NMIIA is crucial to amoeboid durotaxis but how NMIIA is recruited to the soft area remains to be further explored. Migrating amoeboid cells commonly form spherical protrusions called blebs at their leading edges, which facilitate cell locomotion (*Schick and Raz, 2022*). Whether these cell front-positioned blebs are involved in mechanosensing and determination of migrating direction also deserves further investigation.

## Materials and methods
### Cell lines and cell culture
The HL-60 cell line (derived from a 36-year-old female with acute promyelocytic leukemia) was purchased form the National Infrastructure of Cell Line Resource (NICR). Human embryonic kidney 293T (HEK 293T) cells were kept by our laboratory. All cells were passed no more than 10 times and were mycoplasma-free. HL-60 cells were cultured in RPMI-1640 medium (Hyclone, SH30809.01) supplemented with 15% fetal bovine serum (FBS; Vistech, VIS00CU3524), 100 U/mL penicillin and 100 μg/mL streptomycin at 37°C with 5% $CO_2$. The cultured cells were diluted once every 2–3 d into a density of 1×10⁶ cells/mL. For differentiation, HL-60 cells were diluted into RPMI-1640 medium containing 1.3% DMSO (VWR Life Science) with an initial density of 1×10⁶ cells/mL. For all the experiments, only cells differentiated for 5–6 d were used. HEK 293T cells were cultured in Dulbecco's modified Eagle medium (DMEM, Corning, 10–013-CRVC) supplemented with 10% FBS, 100 U/mL penicillin, and 100 μg/mL streptomycin at 37°C with 5% $CO_2$. For cell passage, cells were washed with phosphate-buffered saline (PBS, Macgene, CC010) and digested with 0.25% trypsin (Macgene, CC012).

*Dictyostelium* cells were maintained in HL5 medium containing G418 (10–40 μg/mL), Hygromycin (50 μg/mL), or both as needed, as described in *Li et al., 2022*.

### Antibodies and reagents
The following antibodies were used in this study: anti-α-tubulin (AC012) from ABclonal, anti-MYH9 (A16923) and anti-MYH10 (A12029) from ABclonal, anti-Paxillin (AB_39952) from BD_Biosciences.

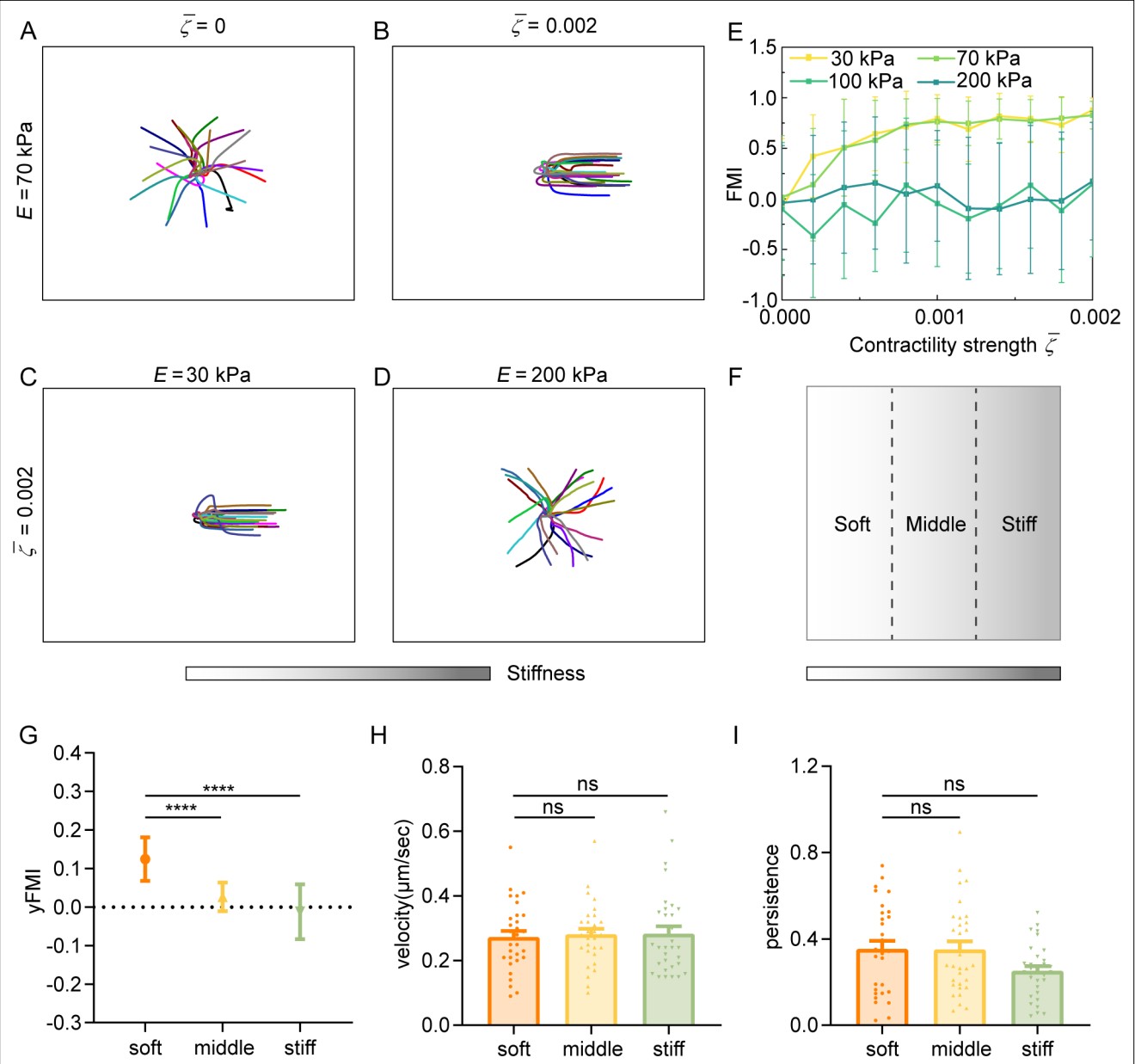

**Figure 6.** Regulation of amoeboid durotaxis by cell contractility and substrate stiffness. (**A**) Trajectories of migrating amoeboid cells with no contractility ($\bar{\zeta}$=0) on the middle region of gradient substrates, and the substrate stiffness of the initial cell centroid was set as $E$=70 kPa. (**B**) Trajectories of migrating amoeboid cells with normal contractility ($\bar{\zeta}$=0.002) on middle region of gradient substrates, and the substrate stiffness of the initial cell centroid was set as $E$=70 kPa. (**C**) Trajectories of migrating amoeboid cells with normal contractility ($\bar{\zeta}$=0.002) on soft region of gradient substrates, and the substrate stiffness of the initial cell centroid was set as $E$=30 kPa. (**D**) Trajectories of migrating amoeboid cells with normal contractility ($\bar{\zeta}$=0.002) on stiff region of gradient substrates, and the substrate stiffness of the initial cell centroid was set as $E$=200 kPa. (**E**) The forward migration index (FMI) with different contractility strength and different initial substrate stiffness of cell centroid. (**F**) Illustration of different regions (soft, middle, and stiff) on gradient substrates with the greater stiffness towards the right. (**G–I**) yFMI, velocity, and migration persistence of CD4[+] Naïve T cells migrating on different regions illustrated in F (n≥30 tracks were analyzed for each experiment, N=3 independent experiments for each condition, replicates are biological). All error bars are SEM. ****p<0.0001, ns, not significant, by one-way ANOVA.

The following reagents were used in this study: Blebbistatin (2946047) from EMD_Millipore, Rho activator II (CN03) and SiR-Actin (CY-SC001) from Cytoskeleton, nocodazole (M1404), CK-666 (182515), and SMIFH2 (S4826) from Sigma-Aldrich.

## NMIIA and NMIIB knockdown cell line generation

The following short hairpin RNA (shRNA) primers were used to generate NMIIA and NMIIB knockdown HL-60 cell lines:

Human shRNA forward sh*MYH9*, 5'- GCCAAGCTCAAGAACAAGCAT-3'.

Human shRNA forward sh*MYH10*, 5'-GCCAACATTGAAACATACCTT-3'.

The plasmid, together with other helper plasmid (pCMV-VSV-G and psPAX2), were transfected to HEK 293T cells at 60–80% confluence. The virus-containing supernatant was harvested 48 hr later, and concentrated 100 folds. $1 \times 10^5$ HL-60 cells were infected with 100 µL concentrated virus supernatant, centrifuging at 1000 g for 2 hr. Polybrene (hexadimethrine bromide) was added at a final concentration of 10 µg/mL to increase infection efficiency. The knockdown efficiency was verified by Western blotting.

## Drug treatment

Drug treatment was performed according to previous publications. CK-666 was used at a concentration of 100 µM and pre-treated for 5 hr before imaging. Blebbistatin was used at a concentration of 10 µM and pre-treated for 10 min before imaging. Rho activator II was used at a concentration of 0.25 µg/mL and pre-treated for 2 hr before imaging. Nocodazole was used at a concentration of 32 µM and pre-treated for 10 min before imaging (*Tabdanov et al., 2021*). Taxol was used at a concentration of 70 nM and pre-treated for 10 min before imaging (*Tabdanov et al., 2021*). SMIFH2 was used at a concentration of 15 µM and pre-treated for 5 hr before imaging. All drugs were incubated for the whole experiment.

## Isolation of murine neutrophils

Bone marrow-derived neutrophils were isolated by Percoll gradients as described by *Li et al., 2019*. Male C57BL/6 mice were euthanized and bone marrow was extracted from the femurs by perfusing 6 mL pre-cold FACS solution (PBS with 2% FBS). Bone marrow cells were pelleted by centrifugation (300 g, 5 min, 4°C) and then resuspended in 2 mL FACS. Percoll solution (Pharmacia) and 10x PBS were mixed (Percoll: 10x PBS = 9:1) to obtain a 100% Percoll stock. 100% Percoll stock was diluted with 1×PBS to get 72%, 60%, and 52% Percoll solutions. Bone marrow cell suspensions were loaded in the uppermost layer. After centrifugation (1100 g, 30 min, 4°C), the neutrophils were collected in the band between the 72% and 60% layers. Before the experiment, the neutrophils were stimulated with 100 nM fMLP (N-formyl-methionyl-leucyl-phenylalanine) (Sigma, F3506) for 10 min.

## CD4+ Naïve T cell purification

Male C57BL/6 mice were euthanized and the spleens were harvested and homogenized with a 40 µm cell strainer. After removing red blood cells by ACK (Ammonium-Chloride-Potassium) lysis, untouched primary naïve T cells were isolated with an EasySep Mouse T cell Isolation Kit according to the manufacturer's protocol (BioLegend). The purified CD4+ Naïve T cells were stimulated with 20 ng/mL phorbol 12-myristate13-acetate (PMA) and 1 µg/mL ionomycin for 4–6 hr to activate the integrin pathway before migration assay. Interaction between Lymphocyte Function-Associated Antigen 1 (LFA-1) and its ligand Intercellular Adhesion Molecule 1 (ICAM-1) has been reported to drive T cell movement by creating tangential friction between the actin cortex and the environment (*Hons et al., 2018*). In this case, T cells were allowed to migrate on the ICAM-1 coated surface.

## Mice

Male C57BL/6 WT mice were obtained from a special pathogen-free facility at Peking University Care Industrial Park, and were used for experiments at the age of 4–6 wk. All procedures were approved by the Animal Care and Use Committee of Peking University and were performed in strict accordance with relevant guidelines and regulations.

## Gradient or uniform PA gel preparation and functionalization

PA-gel preparation and functionalization have been adapted from the previous work in our laboratory (*Rong et al., 2021*; *Wang et al., 2021*). Specially, 40% acrylamide and 2% bis-acrylamide solutions were combined at 11.73% and 0.33%, respectively, to make a 55 kPa gel. 0.5% (w/v) Irgacure 2959 was added to catalyze cross-linking when exposed to UV light. Gels were cast on cover glasses which was treated with bind-silane (BioDee, Beijing, China) followed by treatment with 0.5% glutaraldehyde (Sangon Biotech, Beijing, China) solution. To ensure complete polymerization, the whole acrylamide mix solution was first exposed to the UV light for 10 min without covering (for the initial nucleation in the polymerization reaction). To make gradient stiffness, an opaque mask was put between the gels and UV light. Then the mask was removed slowly by a microinjection pump with constant speed (40 μm/s for 10 min) to produce the steep stiffness gradient gels. After polymerization, the upper coverslip was peeled off the gel, leaving a thin layer of gel on the activated surface. The PA-gel was then washed extensively in PBS. The PA-gel was functionalized with 2 mL soak solution (137 mM NaCl and 5% (v/v) glycerol) for 1 hr at room temperature. After that, 2 mL mixed buffer (pH = 4.5) containing 15 mM 1-ethyl-3-[3-dimethylaminopropyl] carbodiimide hydrochloride (EDC), 25 mM N-hydroxysulfosuccinimide (NHS), 100 mM 2-(N-morpholino) ethane sulfonic acid (MES) and 10% (v/v) glycerol were added. The PA-gel was incubated within the mixed buffer in the dark at room temperature for 30 min. Then, the PA-gel was coated with fibronectin (50 μg/mL) or ICAM-1 (10 μg/mL) diluted in PBS for 35 min at room temperature. The PA-gel was then washed with PBS three times and stored at 4°C for up to 1 wk. The stiffness was measured with atomic force microscopy (AFM).

Uniform PA-gels were made from 40% acrylamide and 2% bis-acrylamide mixed with 10% ammonium persulfate and 1% TEMED, where varying ratios of acrylamide and bis-acrylamide were used to create gels of known reproducible stiffness (*Fischer et al., 2012*). The stiffness was measured with AFM.

## 2D confinement assay

Cells were seeded on the functionalized gradient gel or uniform gel and settled down for about 30 min. The culture medium was aspired and a small amount of medium was left on the gel to prevent detachment and desiccation of the cells. Another gradient or uniform gel was carefully placed on top of the cells to construct a sandwich-like confinement device. This confinement device was incubated at 37°C for at least 30 min allowing the cells to adapt to increased stress in the physical confinement before imaging. To avoid edge effects, fields close to the edge of the gel were not selected for imaging. Unless otherwise stated, all durotaxis assays were proceeded with this confinement device.

## Immunofluorescence and imaging analysis

Cells were plated on acid-washed coverslips coated with 10 μg/mL fibronectin at room temperature for 30 min. Cells were then fixed with 4% paraformaldehyde at room temperature for 15 min, permeabilized with 0.5% Triton X-100 in PBS for 5 min, washed with PBS once for 5 min, and sequentially blocked with 10% bovine serum albumin (BSA) for 1 hr at room temperature. Then, the primary antibody was appropriately diluted in PBS and cells were incubated for 1 hr at room temperature. After washing with PBS three times, the cells were incubated with Alexa Fluor 488- or Alexa Fluor 555-conjugated secondary antibody (Thermo Fisher Scientific) for 1 hr at room temperature. After washing with PBS three times, the coverslips were mounted with Prolong Diamond Antifade (Thermo Fisher Scientific). After the mounting medium was solidified, single-plane images were captured using a 63×1.4 NA objective lens on the Andor Dragonfly epi-fluorescence confocal imaging system.

## Ultrastructure expansion microscopy (U-ExM)

U-ExM procedure was performed as described by *Gambarotto et al., 2021*. Briefly, CD4[+] Naïve T cells were seeded on a gradient PA-gel with another upper gel providing confinement. 4% PFA was used to fix cells for 15 min at room temperature. After fixation, the upper gradient PA-gel is carefully removed and the bottom gradient PA-gel with seeded cells were immersed in an anchoring solution containing 1% acrylamide and 0.7% formaldehyde (Sigma, F8775) for 5 hr at 37 °C. Next, each gradient PA-gel was placed on top of one drop of gelling solution (0.1% N,N'-methylenebisacrylamide, 0.5% ammonium persulfate, 0.5% N,N,N',N'-TEMED, 10% acrylamide and 23% (w/v) sodium acrylate) with cells facing down. The gelling process was fully initiated after 5 min on ice and

completed after 1 hr at 37 °C. The solidified gels were then detached from the gradient PA-gels and denatured in denaturation buffer (50 mM Tris-BASE, 200 mM NaCl, and 200 mM SDS, pH=9.0) for 20 min at 90 °C. In the first round of expansion, the solidified gels were immersed in ddH$_2$O overnight and then labeled with primary and secondary antibodies. Before confocal imaging, the solidified gels were completely expanded through one more round of expansion in ddH$_2$O and then mounted on poly-L-lysine-coated confocal dishes.

## Live cell imaging and durotaxis analysis

Time-lapse videos were acquired for 15 min at 15 s intervals using a 10x dry objective lens (NA 0.3, Olympus, Tokyo, Japan) on a deconvolution microscope (CellTIRF). Images were analyzed with 'manual tracking' and 'chemotaxis tool' plugins of ImageJ. Coordinates and distances of cell movement were tracked using the 'Manual Tracking' plugin of ImageJ, and then tracking data were imported into the 'Chemotaxis Tool' plugin to generate statistic features such as velocity, FMI (forward migration index), directionality, and plot feature such as rose plot.

## Actin flow analysis

Quantitative fluorescence speckle microscopy (qFSM) was applied to analysis the actin flow of dHL-60 cells on PA-gel. In general, dHL-60 cells pretreated with 0.1 µM SiR-Actin for 30 min were plated on PA-gel as described above. Live cell Videos were obtained for 10 min using a 63×1.4 NA objective lens on the Andor Dragonfly confocal imaging system. Videos were analyzed using a previously described qFSM software in MATLAB (*Mendoza et al., 2012*). Full cell masks were generated using automatic thresholding (MinMax setting). Flow analyses were performed using the flow tracking setting with a six-frame integration window. The average actin flow rate was calculated for each Video (n=10–20 cells per group). To compare the overall fluorescence of the cells used in qFSM, an ROI encompassing the cell perimeter was made and the mean fluorescence intensity of this ROI was recorded for each cell.

## Western blotting

For Western blotting, suspended HL-60 cells were harvested and centrifuged at 300 g for 5 min and then washed with PBS once. Cells were lysed in radio immunoprecipitation assay (RIPA) buffer (50 mM Tris-HCl pH 8.0, 150 mM NaCl, 1% Triton X-100, 0.5% Na-deoxycholate, 0.1% SDS, 1 mM EDTA, and protease inhibitor cocktail) for 10 min on ice. Lysates were centrifuged at 12,000 rpm for 10 min at 4°C, and the supernatants were collected. 5x SDS loading buffer was added to the supernatants and the mixture was boiled for 10 min at 95 °C. Protein samples were run on 10% SDS PAGE (H2O, 30% acrylamide, 1.5 M Tris-HCl pH 8.8, 10% ammonium persulphate, and tetramethylethylenediamine (TEMED)) and transferred onto nitrocellulose membranes by wet electrophoretic transfer, followed by 10% nonfat-milk blocking at room temperature for 1 hr. The primary antibody incubations were at 4 °C overnight or at room temperature for 2 hr, followed by the secondary antibody incubations at room temperature for 1 hr.

## AFM measurement

A Bioscope Resolve AFM (AFM, NT-MDT) mounted on an Olympus IX73 inverted microscope was used to investigate the mechanical properties of polyacrylamide gels made from different formulations. The inverted microscope was used to visually position the AFM tip with respect to the specimen. Silicon nitride probes with a squared pyramid tip (DNP, nominal cantilever spring constant=0.06 N/m, Bruker) were used in this study. After system stabilization, the thermal tune method was used to determine the exact spring constant before each experiment. The Hertz-Sneddon model for pyramid tips was used to extract Young's modulus of the sample. For each indentation, the following parameters were used: 1 nN set point, 4.0 µm Z range, and an extend speed of 2.0 µm/s.

## Active gel model

Total hydrodynamic stress $\sigma^{\text{total}}$ includes four parts, including viscous stress:

$$\sigma^{\text{vis}}=\eta \left( \nabla \mathbf{v}+(\nabla \mathbf{v})^{\text{T}}\right), \tag{7}$$

Elastic stress $\sigma^{\text{ela}}$, interfacial stress between active and passive phases $\sigma^{int}$, and active stress $\sigma^{\text{act}}$ (*Hatwalne et al., 2004*). $\eta$ is the shear viscosity, and the elastic stress $\sigma^{\text{ela}}$ is formulated as:

$$\sigma^{\text{ela}} = -\frac{1}{2}\left(\mathbf{P}\frac{\delta F}{\delta \mathbf{P}} - \frac{\delta F}{\delta \mathbf{P}}\mathbf{P}\right) + \frac{\xi}{2}\left(\mathbf{P}\frac{\delta F}{\delta \mathbf{P}} + \frac{\delta F}{\delta \mathbf{P}}\mathbf{P}\right) - \kappa\nabla\mathbf{P}\cdot\left(\nabla\mathbf{P}\right)^{\text{T}}, \tag{8}$$

where $\xi$ is a shape factor. We take $\xi=1.1$, which means shear-aligning rod-like actin are considered (*Kung et al., 2006*). The interfacial stress between the active and passive phase is calculated as:

$$\sigma^{int} = \left(f - c\frac{\delta F}{\delta c}\right)\mathbf{I} - \nabla c\frac{\partial f}{\partial\left(\nabla c\right)}, \tag{9}$$

where $\mathbf{I}$ is the identity tensor and $f$ denotes the free energy density, which has been given by the integrand in *Equation 1*.

The integration is performed using a vorticity or stream-function finite difference scheme (*Giomi et al., 2014*) on a square $128 \times 128$ domain with periodic boundary conditions, whose grid spacing is $\Delta x = \Delta y = 1$. The initial circular droplet with a radius $R = 7$ is placed at the center of the square domain. At the beginning, the unit polarity field $\mathbf{P}$ is randomly generated and the fluid velocity $\mathbf{v}$ is zero. A fourth-order Runge-Kutta method with a time step $\Delta t = 0.1$ is employed to implement the time integration. To normalize the system, three scales are chosen as: length scale $L = 1\,\mu\text{m}$, time scale $T = 10\,\text{ms}$, and force scale $F = 100\,\text{nN}$. The parameters used in the simulations are chosen from previous work (*Tjhung et al., 2012*) and our experiments, which are provided in *Supplementary file 1*.

## Statistical analysis

All the plots and statistical tests were performed using GraphPad Prism software. The presented values were displayed as mean ± SEM for at least three independent experiments (as indicated in the figure legends). The significance was calculated using unpaired Student's $t$-tests or one-way ANOVA, as indicated. In all statistical tests, $p>0.05$ was considered as not significant.

## Acknowledgements

We thank Prof. Dan Lv, Dr. Yuxing Huang, Dr. Yingxue Rong, Dr. Yuan Jin, Dr. Peng Shi, Dr. Chunlei Zhang, and Dr. Xiaoyu Ren from Peking University for technical support and data discussion. We thank Prof. Xiangdong Li from the Chinese Academy of Sciences for sharing the NMIIA construct. Funding: National Key Research and Development Program of China 2022YFC3401100 (CW) National Natural Science Foundation of China grant 32122029 (CW) National Natural Science Foundation of China grant 11922207 (BL).

## Additional information

### Funding

| Funder | Grant reference number | Author |
| --- | --- | --- |
| National Key Research and Development Program of China | 2022YFC3401100 | Congying Wu |
| National Natural Science Foundation of China | 32122029 | Congying Wu |
| National Natural Science Foundation of China | 11922207 | Bo Li |

The funders had no role in study design, data collection and interpretation, or the decision to submit the work for publication.

## Author contributions
Chenlu Kang, Xin Yi, Conceptualization, Formal analysis, Investigation, Visualization, Methodology, Writing - original draft, Writing - review and editing; Pengcheng Chen, Software, Formal analysis, Investigation, Visualization, Methodology, Writing - original draft, Writing - review and editing; Dong Li, Yihong Yang, Methodology; Yiping Hu, Writing - review and editing; Huaqing Cai, Supervision, Methodology, Project administration; Bo Li, Congying Wu, Conceptualization, Supervision, Funding acquisition, Methodology, Project administration, Writing - review and editing

## Author ORCIDs
Huaqing Cai ![ORCID] https://orcid.org/0000-0001-8434-6639
Congying Wu ![ORCID] https://orcid.org/0000-0003-0007-0204

## Ethics
In this study, all animal experiments were approved by the Animal Care and Use Committee of Peking University (Permit Number: LA2020398) and were performed in strict accordance with relevant guidelines and regulations.

Reviewer #1 (Public review): https://doi.org/10.7554/eLife.96821.4.sa1
Reviewer #2 (Public review): https://doi.org/10.7554/eLife.96821.4.sa2
Author response https://doi.org/10.7554/eLife.96821.4.sa3

# Additional files

## Supplementary files
• Supplementary file 1. Parameters used in simulations.

• MDAR checklist

## Data availability
All imaging data, plot data, Western data and code associated with our work have been submitted to Dryad.

The following dataset was generated:

| Author(s) | Year | Dataset title | Dataset URL | Database and Identifier |
|---|---|---|---|---|
| Wu C | 2024 | Data from: Amoeboid cells undergo durotaxis with soft end polarized NMIIA | http://dx.doi.org/10.5061/dryad.8cz8w9h1x | Dryad Digital Repository, 10.5061/dryad.8cz8w9h1x |

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
