## [Editor Report · eLife Assessment]

This study presents an **important** finding on durotaxis in various amoeboid cells that is independent of focal adhesions. The evidence supporting the authors' claims is **compelling**. The work will be of interest to cell biologists and biophysicists working on rigidity sensing, the cytoskeleton, and cell migration.

---

## [Referee Report · Reviewer #1 (Public review)]

In their paper, Kang et al. investigate rigidity sensing in amoeboid cells, showing that, despite their lack of proper focal adhesions, amoeboid migration of single cells is impacted by substrate rigidity. In fact, many different amoeboid cell types can durotax, meaning that they preferentially move towards the stiffer side of a rigidity gradient.

The authors observed that NMIIA is required for durotaxis and, buiding on this observation, they generated a model to explain how durotaxis could be achieved in the absence of strong adhesions. According to the model, substrate stiffness alters the diffusion rate of NMAII, with softer substrates allowing for faster diffusion. This allows for NMAII accumulation at the back, which, in turn, results in durotaxis.

The evidence provided for durotaxis of non adherent (or low-adhering) cells is strong. I am particularly impressed by the fact that amoeboid cells can durotax even when not confined. I wish to congratulate the authors for the excellent work, which will fuel discussion in the field of cell adhesion and migration.

---

## [Referee Report · Reviewer #2 (Public review)]

Summary:

The authors developed an imaging-based device, that provides both spatial confinement and stiffness gradient, to investigate if and how amoeboid cells, including T cells, neutrophils and *Dictyostelium* can durotax. Furthermore, the authors showed that the mechanism for the directional migration of T cells and neutrophils depends on non-muscle myosin IIA (NMIIA) polarized towards the soft-matrix-side. Finally, they developed a mathematical model of an active gel that captures the behavior of the cells described in vitro.

Strengths:

The topic is intriguing as durotaxis is essentially thought to be a direct consequence of mechanosensing at focal adhesions. To the best of my knowledge, this is the first report on amoeboid cells that are not dependent on FAs to exert durotaxis. The authors developed an imaging-based durotaxis device that provides both spatial confinement and stiffness gradient and they also utilized several techniques such as quantitative fluorescent speckle microscopy and expansion microscopy. The results of this study have well-designed control experiments and are therefore convincing.

---

## [Author Response]

The following is the authors’ response to the previous reviews.

**Public Reviews:**

**Reviewer #1 (Public Review):**
In their paper, Kang et al. investigate rigidity sensing in amoeboid cells, showing that, despite their lack of proper focal adhesions, amoeboid migration of single cells is impacted by substrate rigidity. In fact, many different amoeboid cell types can durotax, meaning that they preferentially move towards the stiffer side of a rigidity gradient.The authors observed that NMIIA is required for durotaxis and, buiding on this observation, they generated a model to explain how durotaxis could be achieved in the absence of strong adhesions. According to the model, substrate stiffness alters the diffusion rate of NMAII, with softer substrates allowing for faster diffusion. This allows for NMAII accumulation at the back, which, in turn, results in durotaxis.The authors responded to all my comments and I have nothing to add. The evidence provided for durotaxis of non adherent (or low-adhering) cells is strong. I am particularly impressed by the fact that amoeboid cells can durotax even when not confined. I wish to congratulate the authors for the excellent work, which will fuel discussion in the field of cell adhesion and migration.

We thank the reviewer for critically evaluating our work and giving kind suggestions. We are glad that the reviewer found our work to be of potential interest to the broad scientific community.

**Reviewer #2 (Public Review):**
Summary:The authors developed an imaging-based device that provides both spatialconfinement and stiffness gradient to investigate if and how amoeboid cells, including T cells, neutrophils, and *Dictyostelium*, can durotax. Furthermore, the authors showed that the mechanism for the directional migration of T cells and neutrophils depends on non-muscle myosin IIA (NMIIA) polarized towards the soft-matrix-side. Finally, they developed a mathematical model of an active gel that captures the behavior of the cells described in vitro.Strengths:The topic is intriguing as durotaxis is essentially thought to be a direct consequence of mechanosensing at focal adhesions. To the best of my knowledge, this is the first report on amoeboid cells that do not depend on FAs to exert durotaxis. The authors developed an imaging-based durotaxis device that provides both spatial confinement and stiffness gradient and they also utilized several techniques such as quantitative fluorescent speckle microscopy and expansion microscopy. The results of this study have well-designed control experiments and are therefore convincing.Weaknesses:Overall this study is well performed but there are still some minor issues I recommend the authors address:(1) When using NMIIA/NMIIB knockdown cell lines to distinguish the role of NMIIA and NMIIB in amoeboid durotaxis, it would be better if the authors took compensatory effects into account.

We thank the reviewer for this suggestion. We have investigated the compensation of myosin in NMIIA and NMIIB KD HL-60 cells using Western blot and added this result in our updated manuscript (Fig. S4B, C). The results showed that the level of NMIIB protein in NMIIA KD cells doubled while there was no compensatory upregulation of NMIIA in NMIIB KD cells. This is consistent with our conclusion that NMIIA rather than NMIIB is responsible for amoeboid durotaxis since in NMIIA KD cells, compensatory upregulation of NMIIB did not rescue the durotaxis-deficient phenotype.

(2) The expansion microscopy assay is not clearly described and some details are missed such as how the assay is performed on cells under confinement.

We thank the reviewer for this comment. We have updated details of the expansion microscopy assay in our revised manuscript in line 481-485 including how the assay is performed on cells under confinement:

Briefly, CD4+ Naïve T cells were seeded on a gradient PA gel with another upper gel providing confinement. 4% PFA was used to fix cells for 15 min at room temperature. After fixation, the upper gradient PA gel is carefully removed and the bottom gradient PA gel with seeded cells were immersed in an anchoring solution containing 1% acrylamide and 0.7% formaldehyde (Sigma, F8775) for 5 h at 37 °C.

(3) In this study, an active gel model was employed to capture experimental observations. Previously, some active nematic models were also considered to describe cell migration, which is controlled by filament contraction. I suggest the authors provide a short discussion on the comparison between the present theory and those prior models.

We thank the reviewer for this suggestion. Active nematic models have been employed to recapitulate many phenomena during cell migration (*Nat Commun.*, 2018, doi: 10.1038/s41467-018-05666-8.). The active nematic model describes the motion of cells using the orientation field, Q, and the velocity field, u. The director field n with (n = −n) is employed to represent the nematic state, which has head-tail symmetry. However, in our experiments, actin filaments are obviously polarized, which polymerize and flow towards the direction of cell migration. Therefore, we choose active gel model which describes polarized actin field during cell migration. In the discussion part, we have provided the comparison between active gel model and motor-clutch model. We have also supplemented a short discussion between the present model and active nematic model in the main text of line 345-347:

The active nematic model employs active extensile or contractile agents to push or pull the fluid along their elongation axis to simulate cells flowing (61).

(4) In the present model, actin flow contributes to cell migration while myosin distribution determines cell polarity. How does this model couple actin and myosin together?

We thank the reviewer for this question. In our model, the polarization field P(r,t) is employed to couple actin and myosin together. It is obvious that actin accumulate at the front while myosin diffuses in the opposite direction. Therefore, we propose that actin and myosin flow towards the opposite direction, which is captured in the convection term of actin ∇⋅[c(v+wP)] and myosin (∇⋅[m(v−wP)]) density field.